# Control of innate olfactory valence by segregated cortical amygdala circuits

James R Howe[1,2,3†], Chung Lung Chan[1†], Donghyung Lee[1], Marlon Blanquart[1], James H Lee[1], Laurine Decoster[1], Haylie K Romero[2,3,4], Abigail N Zadina[5‡], Mackenzie E Lemieux[6], Fergil Mills[6§], Paula A Desplats[3,4,7], Kay M Tye[1,6,8], Cory M Root[1*]

[1]Department of Neurobiology, University of California, San Diego, La Jolla, United States; [2]Neurosciences Graduate Program, University of California, San Diego, La Jolla, United States; [3]Department of Neurosciences, University of California, San Diego, La Jolla, United States; [4]Center for Circadian Biology, University of California, San Diego, La Jolla, United States; [5]Zuckerman Mind Brain Behavior Institute, Columbia University, New York, United States; [6]Salk Institute for Biological Sciences, La Jolla, United States; [7]Department of Pathology, University of California, San Diego, La Jolla, United States; [8]Howard Hughes Medical Institute, La Jolla, United States

*For correspondence: cmroot@ucsd.edu

†These authors contributed equally to this work

Present address: ‡Department of Neuroscience and Behavior, Barnard College of Columbia University, New York, United States; §Department of Neurobiology, University of Utah School of Medicine, Salt Lake, United States

Competing interest: The authors declare that no competing interests exist.

## eLife Assessment

This **fundamental** manuscript describes how the posterolateral cortical amygdala (plCoA) generates appetitive or aversive behaviors in response to odors. By combining optogenetic stimulation, single-cell RNA sequencing, and spatial analysis, the authors identify a topographically organized circuit within plCoA that governs these behaviors. The manuscript shows **convincingly** that multiple features (spatial, genetic, and projection) contribute to overall population encoding of valence. Overall, the authors conduct many challenging experiments, each of which contains the relevant controls, and the results are interpreted within the framework of their experiments.

**Abstract** Animals exhibit innate behaviors that are stereotyped responses to specific evolutionarily relevant stimuli in the absence of prior learning or experience. The posterolateral cortical amygdala (plCoA) mediates innate attraction and aversion to odor. Here, we sought to define the circuit features of plCoA that give rise to innate attraction and aversion to odor in mice. First, we examined odor-evoked responses in these areas and found sparse encoding of odor identity, but not valence. We next considered a topographic organization and found that optogenetic stimulation of the anterior and posterior domains of plCoA elicits avoidance and attraction, respectively, suggesting a functional axis for valence. Using single-cell and spatial RNA sequencing, we identified the molecular cell types in plCoA, revealing an anteroposterior gradient in glutamatergic neurons that are sufficient and partially necessary for behavior. Finally, we identified topographically organized projections, whereby anterior neurons preferentially project to medial amygdala, and posterior neurons preferentially project to nucleus accumbens, which are respectively sufficient and necessary for innate attraction and aversion. Together, these data support a model whereby distinct, topographically distributed plCoA populations direct innate olfactory responses by signaling to divergent valence-specific targets, linking upstream olfactory identity to downstream valence behaviors, through a population code.

## Introduction

Innate behaviors are ubiquitous across the animal kingdom, allowing specific sensory stimuli to yield stereotypical behavioral responses even in the absence of learning or past experience. These behaviors include feeding, fighting, fleeing, and mating, among others, and many can be simplified onto an axis of positive or negative valence representing approach and avoidance. Innate behaviors are the result of evolutionary selection, guiding initial behaviors that can be updated by future experiences. Given that innate behaviors are genetically hardwired, it is thought they should be mediated by simple circuits with specified connections between layers of the nervous system. Innate behaviors are common across sensory modalities but are especially prominent in olfaction, whereby diverse chemical signals, critical to survival and reproduction, must be quickly and robustly detected and processed, in the absence of prior experience. For instance, predator odors represent a potentially imminent threat and necessitate a quick, decisive, aversive response (*Stowers et al., 2013*). Conversely, innately appetitive odors represent potentially rewarding stimuli like food or heterospecifics, inducing attraction (*Li and Liberles, 2015*). These odors comprise a small subset of perceptible chemical space, and the detection of specific odorants is both species-specific and under genetic control (*Hayden et al., 2010*; *Saraiva et al., 2019*; *Ibarra-Soria et al., 2017*).

Valence is a fundamental perceptual feature of olfaction (*Yeshurun and Sobel, 2010*). Motivational valence can be defined as seeking or avoiding specific stimuli, and it is observed across sensory stimuli in both innate and learned responses. Multiple circuit motifs have been proposed to mediate such valence responses (*Tye, 2018*). In the simplest form, labeled line motifs segregate information from sensation to action throughout the nervous system. This has been observed in the taste and somatosensory systems, as well as hints of labeled lines in the olfactory system, where individual glomeruli are necessary and sufficient for innate responses consistent with this model (*Dewan et al., 2013*; *Saito et al., 2017*; *Semmelhack and Wang, 2009*). Many circuits are organized along divergent path motifs, where a region receives the same sensory input but instead acts akin to a switchboard, processing and sorting its output to distinct downstream targets to convey positive or negative signals. This motif is most associated with the BLA and associated learning processes (*Tye, 2018*; *Beyeler et al., 2018*). Still other circuits contain opposing components motifs, in which opposing inputs target a single effector region to control the balance of one target. It remains unclear which, if any, of these generalized circuit motifs are present in the olfactory system.

Olfactory sensation begins with olfactory sensory neurons (OSNs) in the olfactory epithelium (OE) that express a single receptor, projecting to spatially stereotyped glomeruli in the olfactory bulb (OB). Postsynaptic mitral/tufted cells within the OB project in parallel to third-order olfactory areas, including the posterolateral cortical amygdala (plCoA). Unlike in other third-order olfactory areas, such as piriform cortex, projections from individual glomeruli from the OB to plCoA are spatially restricted and stereotyped, consistent with genetically hardwired circuits (*Sosulski et al., 2011*; *Miyamichi et al., 2011*). Past work has demonstrated that plCoA is necessary and sufficient for innate olfactory responses, with spatially ordered labeling of responsive neurons via immediate early gene labeling (*Root et al., 2014*). However, other work suggests that there is no spatial organization to odor responses or valence encoding in plCoA (*Iurilli and Datta, 2017*). Thus, a further investigation of plCoA organization is necessary to understand how this structure controls innate olfactory responses.

Distinct brain areas employ distinct coding strategies to represent information. Neuronal ensembles within all other major olfactory regions observed thus far, such as the anterior olfactory nucleus, OB, OE, olfactory tubercle (OT), piriform cortex (PIR), and tenia tecta, all generally perform sparse population encoding of odor identity, despite the major differences in neuronal composition, organization, and function between the six regions (*Iurilli and Datta, 2017*; *Tsuji et al., 2019*; *Malnic et al., 1999*; *Stettler and Axel, 2009*; *Soucy et al., 2009*; *Lee et al., 2023*; *Cousens, 2020*). On the other hand, ensembles within amygdala subnuclei, most notably the basolateral amygdala (BLA), instead tend to represent the valence of stimuli, with considerable heterogeneity based on a given population's projection target, molecular identity, and topography (*Beyeler et al., 2018*; *Kim et al., 2016*). A complete investigation of the encoding properties and organization of plCoA is needed to determine how innate olfactory valence emerges from this region.

To identify the plCoA circuitry that underlies innate attraction and avoidance to odor, we investigated multiple intersecting scales of organization, from single cell transcriptomes and spatial gene expression, to circuit mapping, manipulation, and physiology. First, we used two-photon calcium

imaging to find that plCoA ensembles encode odor identity, but not valence. Next, we identify a functional gradient in plCoA where activation of anterior and posterior neurons drives responses of opposing valence. Next, we characterized the cell types within plCoA, identifying novel, molecularly defined populations specific to each domain of plCoA, which are respectively sufficient and partially necessary for innate olfactory valence. Further, we perform comprehensive projection mapping to identify downstream projection targets of plCoA, identifying projections to the medial amygdala (MeA) and nucleus accumbens (NAc) that are enriched based on molecular and topographic identity. Finally, manipulations of neuronal ensembles projecting to these targets are sufficient and necessary to control innate olfactory valence responses. Together, these findings identify a novel topographically distributed circuit from plCoA to MeA and NAc that controls innate olfactory aversion and attraction, respectively, consistent with a hybrid model mixing features of labeled lines and divergent path circuit motifs.

## Results

### Population encoding of odor identity in plCoA

To better understand how plCoA circuitry mediates innate attraction and aversion, we first decided to examine the relationship between its spatial organization and odor-evoked activity. Prior analysis of immediate early gene expression following minutes-long odor exposure suggests that activity in the anterior and posterior domains of plCoA could respectively mediate innate aversive and appetitive odor responses (*Root et al., 2014*). However, in vivo electrophysiology with high-density electrode arrays found no evidence for spatial organization or valence encoding in plCoA (*Iurilli and Datta, 2017*). These two studies propose contradictory models of plCoA encoding properties that are both plausible, given that spatial organization is common in the extended amygdala and insular cortex (*Chen et al., 2011*), and population coding is ubiquitous across olfactory regions (*Stettler and Axel, 2009*). We speculated that technical differences could be responsible for these opposing findings. Immediate early gene labeling has low temporal resolution and likely requires a high amount of neural activity to activate gene expression. On the other hand, the recording sites in the latter study appear biased towards the middle of plCoA, and odor was given for 2 s in interleaved trials, whereas attraction and avoidance responses have been measured on a minutes-long timescale (*Root et al., 2014*; *Kobayakawa et al., 2007*). Although it is unclear when the valence of an odor is first perceived, we wondered if the apparent contradictions in these studies could be resolved by applying a longer odor delivery protocol that better matches the timescale of behavioral readouts and balances spatial and temporal resolution.

Therefore, we developed an approach to image neural activity in plCoA with a modified odor delivery schedule, expressing GCaMP8s targeted towards either the anterior or posterior subsection of plCoA and implanting a gradient-index relay (GRIN) lens above to allow in vivo imaging of calcium transients via head-fixed two-photon microscopy (*Figure 1A–C*, *Figure 1—figure supplement 1A-B*). We then examined calcium responses in these mice during a long odor exposure, where odors were presented repeatedly in 20-trial blocks for 5 s each in counterbalanced order (*Figure 1D*). We chose two odorants of each innate valence: the appetitive odors 2-phenylethanol (2PE) and peanut oil (Peanut), the neutral odors heptanol (HEP) and isoamyl acetate (IAA), and the aversive odors trimethylthiazoline (TMT) and 4-methylthiazoline (4MT; *Root et al., 2014*). In total, we recorded Ca$^{2+}$ signals from 345 neurons across 13 mice.

First, we pooled anterior and posterior plCoA neurons together and performed hierarchical clustering of their trial-averaged responses to the 6 odors to categorize odor responses in an unbiased manner (*Figure 1E*). Consistent with the previous in vivo electrophysiology study, we found that the majority of plCoA neurons did not seem to selectively respond to odors of one valence group (*Figure 1E–F*). Across mice, a majority of plCoA neurons did not reliably respond to any of the six odors, and activity was sparse: only 34.5% of plCoA neurons responded to 1 of the 6 odors and 10%–2 odors, while a much smaller portion of plCoA neurons responded to 3 or more odors (*Figure 1G*). Further, we found no significant relationship between the valence of odor and proportion of responsive neurons, and no difference in the proportion responsive to the different odors across anterior and posterior plCoA, suggesting a lack of bias in responsiveness to aversive or appetitive odors across the anterior-posterior axis (*Figure 1H*).

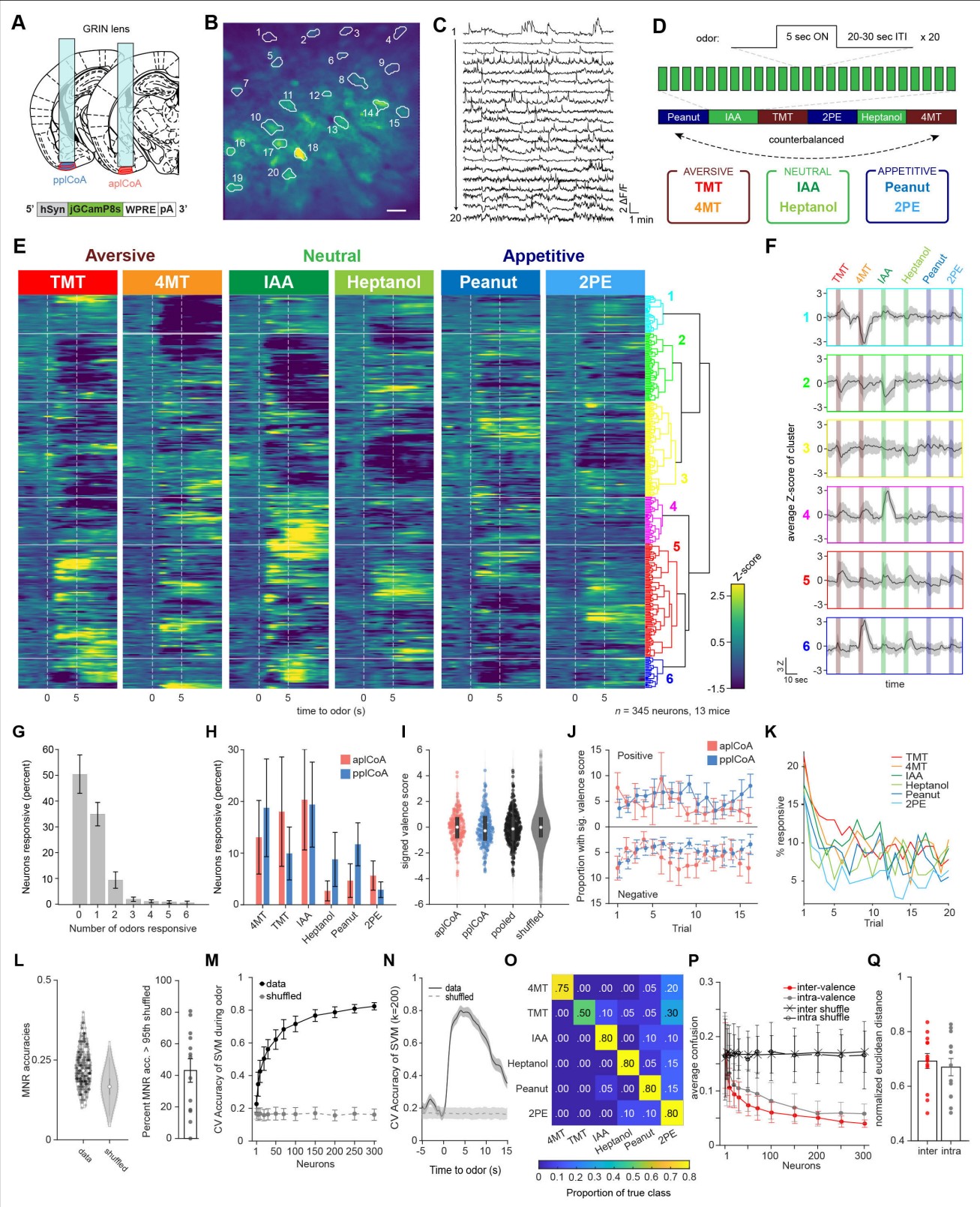

**Figure 1.** The plCoA encodes odors of innate-valence using a population code for odor identity. (**A**) Schematic representation of virus injection and GRIN lens implantation into aplCoA or pplCoA for two-photon microscopy. (**B–C**) Representative images (**B**) and traces (**C**) of fluorescence changes in individual neurons over an approximately 20-min period that includes periods of odor stimulation. Scale bar in (**B**), 100μm. (**D**) Schematic of odor exposure paradigm. Each trial presented 5 seconds of odor followed by a variable inter-trial interval (20–30 s). Odors were present in blocks of 20 trials

*Figure 1 continued on next page*

*Figure 1 continued*

per odor, with 2 counterbalanced block schedules (1 and 2). Six odors were used: the appetitive odors 2-phenylethanol (2PE) and peanut oil (Peanut), the neutral odors heptanol (HEP) and isoamyl acetate (IAA), and the aversive odors trimethylthiazoline (TMT) and 4-methylthiazoline (4MT). (**E**) Heatmap of trial-averaged and Z-scored odor-evoked activity over time from pooled plCoA neurons. Responses are grouped by hierarchical clustering, with the dendrogram (right). Odor delivery marked by vertical red lines. (**F**) Average of trial-averaged and Z-scored odor-evoked activity for each cluster concatenated. The order of color-coded blocks corresponds to the order of clusters in (**E**). (**G**) Proportion of neurons responsive to different numbers of odors. Bars represent the mean across 13 animals, and the error bars show SEM. (**H**) Proportion responsive to each odor for aplCoA (red) or pplCoA (blue). (**I**) Valence scores of individual neurons. White circles show the median of each distribution, whereas the gray rectangle shows the 25th-75th percentile range. (**J**) Proportion of neurons with significant valence scores calculated as a function of trial number. Calculated with a 10-trial moving window. Top half shows those with significant positive valence scores, the bottom half shows those with significant negative valence scores. (**K**) The percentage of neurons with responses (Z>2 for at least 5 frames) as a function of trial number for each odor. (**L**) Left, MNR accuracies for all pooled plCoA neurons (data) and a control distribution where the training labels are shuffled (shuffled) in a violin plot. Right, proportion of neurons in each animal that have MNR accuracy greater than the 95th percentile of the shuffled MNRs. (**M**) Cross-validated average accuracies of multinomial SVMs plotted as a function of the number of neurons used for training during the odor period. Circles represent the mean across 100 iterations of random sampling of neurons and error bars show the standard deviation. (**N**) Cross-validated accuracy of ecoc-SVM classifiers for a six-odor classification task trained using 200 neurons as a function of time. Lines indicate means and shaded areas show the standard deviation across 100 random samplings of 200 neurons from the pooled data and shuffled training controls where the label vectors are randomly shuffled. (**O**) An example confusion matrix for a multinomial SVM trained with 200 neurons. (**O**) Comparison of inter-valence and intra-valence confusion across number of neurons used in training the classifiers. Filled circles show the average of the data across 100 iterations, open circles show shuffled controls. (**Q**) The normalized average distance between odor pairs that have different valence (inter) or same valence (intra). Across panels, ns, not significant. Additional specific details of statistical tests can be found in ***Supplementary file 1***.

The online version of this article includes the following figure supplement(s) for figure 1:

**Figure supplement 1.** Additional information for imaging experiments.

**Figure supplement 2.** Analysis of odor-evoked calcium activity with respect to walking behavior during imaging trials.

We next attempted to quantify valence encoding by calculating a valence score for each neuron by subtracting the average integrated aversive odor response from the average integrated appetitive odor response and dividing this resulting difference in valence by the integrated odor response ($V = \frac{\Sigma_{appetitive} - \Sigma_{aversive}}{\Sigma_{total}}$). Although the valence score was more negatively skewed amongst anterior neurons than posterior ones, there was no significant difference in their distributions, indicating that anterior and posterior plCoA neurons do not broadly encode opposing appetitive and aversive responses (***Figure 1I***). Given that the innate aversive and appetitive behaviors evolve over time in freely moving animals, we wanted to assess if odor responses changed between early and late trials. Using a moving window of five trials, we examined if the proportion of neurons with significant aversive or appetitive valence scores differed over time in anterior or posterior plCoA but found no significant interaction between time and imaging location (***Figure 1J***).

Repeated olfactory stimuli, with the same odor, were used to assess potential changes in odor representation that evolve over a time scale more consistent with our behavioral assay, but this raises concerns about stability of odor delivery across trials and neuronal habituation. To address this, we first assessed the reliability of odor delivery and neuronal responses across trials. PID (Photo Ionization Detector) recordings of odor stimuli across trials revealed moderately stable delivery of all odorants (***Figure 1—figure supplement 1C and D***). Neutral odors were the least stable with a 29–39% decrease in PID detection between the first and last five trials, whereas the attractive and aversive odors decreased by 15–18%. Although the odors do not entirely deplete over time, these changes in odor concentration may affect physiological responses to odor. Next, neuronal response reliability was assessed by examining the frequency of neurons responding with a z score greater than 2 to each odor. With this criterion, the majority of neurons responded to less than half of the trials, while a subset responded to as many as 17/20 trials (***Figure 1—figure supplement 1E and F***). Using a less stringent criteria for responsiveness (z score >1), more neurons appear to respond on more than half of the trials, with a minority responding to all trials (not shown). It is noteworthy that these response probabilities are lower than those observed by Iurilli et al., using intermingled odor stimuli (***Iurilli and Datta, 2017***), indicating that repeated stimulus appears to cause some habituation in odor response, which may also be partly due to a decrease in odor concentration. However, despite the weak reliability of individual neurons, the total number of responsive neurons in the population remained fairly consistent across trials (approximately 5–10%) after a decrease following the first two trials (***Figure 1K***). Thus, despite some changes in representation across trials, we do not see the

emergence of valence, and this doesn't appear to result from failure to deliver odor or stimulate neurons.

We further investigated odor encoding at the single neuron level using multinomial regression (MNR) six-odor classifiers trained on single neuron data and found that they perform only marginally better than chance, suggesting that the majority of single plCoA neurons do not broadly encode discriminatory information about odor identity (*Figure 1K*). Across mice, only 43.1% have overall classification rates above the 95th percentile of shuffled controls, indicating that the majority of single plCoA neurons do not broadly encode discriminatory information about odor identity. Interestingly, when comparing the ranked sub-accuracies for single odors compared to shuffled controls, we found the discrepancy between data and shuffled controls to be accentuated in the two highest performing sub-accuracies (*Figure 1—figure supplement 1G*). Quantifying the proportion of neurons with sub-accuracies higher than the 95th percentile of the shuffled across biological replicates, we found that the decrease of sub-accuracies down the rank falls exponentially, rather than linearly (*Figure 1—figure supplement 1H*). This observation suggests that the individual neurons examined here have little information about the odors tested. This may seem surprising given that sparsely responsive neurons should carry highly specific information about odor identity. However, it is important to note that we are under sampling the large stimulus space (likely billions of odors), thus the neurons we sampled seem to carry little information about the limited stimulus space we tested. It is also possible that faint reliability dilutes the averaged signal used to decode from single neurons. Nonetheless, we do not find an enrichment for neurons specific to odors of valence.

Since we did not observe any evidence of valence encoding at the single neuron level, we next considered population level encoding. The plCoA neurons appear to primarily encode odor identity in a sparse manner, much like the other higher-order olfactory areas, known to use a population code. In contrast to the poor single neuron classification, SVMs trained on population-level data vastly outperform the shuffled data, indicating good encoding of identity at the population level (*Figure 1M and N*). We further used a confusion matrix to ask whether training the classifier with a given odor could accurately predict the identity of another odor. In the matrix, each row corresponds to a predicted class and each column corresponds to the actual class (*Figure 1O*). If the population level activity is similar between odor pairs, we expect the confusion to be higher than less similar pairs. The output of a confusion matrix yields a proportion of true class labels, whereby a high proportion indicates good prediction. We found no difference between the confusion rates for intra-valence classification or inter-valence classification, suggesting a lack of valence encoding at the population level (*Figure 1P*). The similarity between inter-valence and intra-valence confusion was mirrored when quantifying the range-normalized pairwise Euclidean distance across biological replicates (*Figure 1Q*, *Figure 1—figure supplement 1I*). Thus, plCoA appears to encode odor identity in a population code like other higher order olfactory regions, with no apparent encoding of valence.

We next examined whether odor tuning varied with odor-evoked behavior during head fixation. Walking velocity was recorded on a treadmill-linked encoder during imaging trials, enabling analysis of odor-evoked locomotor responses and trial sorting based on behavioral patterns. To classify behavioral patterns, normalized walking velocity across trials was pooled and subjected to hierarchical clustering (*Figure 1—figure supplement 2A*). This analysis revealed distinct odor-evoked locomotor responses with different temporal profiles (*Figure 1—figure supplement 2B*). For example, cluster I displayed increased velocity early during the odor that continued for ~5 s after odor delivery, whereas cluster III displayed increased velocity beginning at odor termination. The majority of trials fell into cluster IV, which showed little change in odor-evoked walking velocity. To determine whether locomotor responses were odor specific, we compared cluster prevalence for each odor (*Figure 1—figure supplement 2C*) and the distribution of odor within clusters as a function of trial number (*Figure 1—figure supplement 2D*). Although we did not observe any reliable valence-specific behavioral responses to odor in head-fixed mice, we did observe divergent patterns of walking behavior unrelated to odor identity. To test whether these clusters of behavioral responses are associated with changes in odor representation, we analyzed neural activity based on its respective behavioral cluster for each trial (*Figure 1—figure supplement 2E and F*). Sorting the data by behavioral response type did not reveal any valence-specific responses but did uncover systematic differences in neural activity. For example, in cluster I, when animals increase velocity at odor onset, we see a general decrease in odor-evoked activity. In contrast, the broadest responses to odors are found in cluster IV, where

animals exhibit little change in behavior. Overall, we did not detect valence-specific behavioral or neural responses to odor in head-fixed mice.

## A functional axis for valence in plCoA

Since we did not observe clear evidence for valence encoding, we considered other organizational principles that could support appetitive and aversive behaviors. Spatial organization for valence processing has been previously observed in the insular cortex and BLA, whereby discrete subsections of the region contain neurons wired to preferentially signal positive or negative valence (*Beyeler et al., 2018*; *Kim et al., 2016*; *Wang et al., 2018b*). Given plCoA's spatially ordered afferent projections from OB, we next hypothesized that plCoA circuitry could still be organized along the anterior-posterior (AP) axis to support attraction and aversion. If true, it follows that activation of small ensembles of neurons along the AP axis should generate behavioral responses along a corresponding axis of valence. We tested this prediction by expressing channelrhodopsin (ChR2) in subsets of neurons at different positions along this axis and photostimulating them during behavioral testing in the previously established four-quadrant open field arena (*Root et al., 2014*).

Based on cytoarchitecture, we parcellated plCoA into three domains: the anterior plCoA (aplCoA), a two-layered region on the ventral surface lateral to the anterior cortical amygdala, the posterior plCoA (pplCoA), a three-layered region on the ventrolateral surface lateral to the posteromedial cortical amygdala, and a middle zone (mplCoA) between them (*Figure 2A*; *Root et al., 2014*). To determine the potential relationship between position on the anterior-posterior axis of plCoA and evoked behavior, we performed optogenetic stimulation at points along this entire axis, expressing channelrhodopsin (ChR2; *Boyden et al., 2005*) and implanting fibers into each zone (*Figure 2B–C*, *Figure 2—figure supplement 1A*). Behavioral response was assessed using the four-quadrant open field assay, where mice freely explored a chamber with or without stimulation, and approach or avoidance was scored with a performance index measuring quadrant occupancy relative to chance, as well as the mean distance to the corner port. Mice were tested for a 10-min baseline period followed by 15 min of closed loop optogenetic stimulation (470 nm, increasing from 1 to 10 Hz as the mouse proceeds closer to the corner port) in one quadrant (*Figure 2D*) as previously done (*Root et al., 2014*).

Throughout the trial period, we observed a negative linear relationship between the anterior-posterior position of photostimulation site and the valence of the behavior. The amount of time spent in the stimulated quadrant as well as the distance to the corner port varied in ChR2, but not eYFP-infected mice, whereby responses shifted from appetitive to aversive as stimulation became more anterior (*Figure 2E–F*). We grouped these responses to determine whether these opposing responses were specific to the identified plCoA zones. We found that photostimulation in aplCoA significantly reduced time spent in the 'on' quadrant and increased the average distance to the corner port during the treatment period, indicating activation of aplCoA neurons is aversive and leads to avoidance of the quadrant paired with stimulation (*Figure 2G–I*). We also found the opposite was true in pplCoA, where stimulation in that zone instead increased the time in the 'on' quadrant and decreased average distance to the corner port, indicating pplCoA neuron activation instead is appetitive and leads to attraction to the stimulation quadrant (*Figure 2J–L*).

In the experiment described above, ChR2 was expressed in thousands of neurons, a scale that is not physiologically representative. This led us to ask whether activating a sparse ensemble of neurons in either the anterior plCoA (aplCoA) or posterior plCoA (pplCoA) would evoke similar behavioral responses. To achieve sparse labeling, we used an immediate early gene-based strategy to label odor-responsive neurons. The Arc-creER$^{T2}$ mouse was previously used in combination with Cre-dependent AAVs to label odor-responsive neurons in plCoA for broad activation (*Root et al., 2014*). We adapted this approach by targeting AAV-DIO-ChR2-eYFP to either the anterior or posterior plCoA followed by tamoxifen induction and TMT exposure. The odor TMT was used as because it labels neurons across the entire plCoA with this system (*Root et al., 2014*). Although TMT is an aversive odor, if posterior plCoA neurons that drive attraction are activated by TMT, then selective activation of this subset without recruiting the full TMT-responsive ensemble should generate an approach response. As established previously, this labeling strategy captures approximately half of Arc-positive neurons (40–100 neurons per 100 μm; *Root et al., 2014*), leading us to estimate that only a few hundred neurons express ChR2. Mice were then assayed for approach or avoidance behavior during photoactivation of

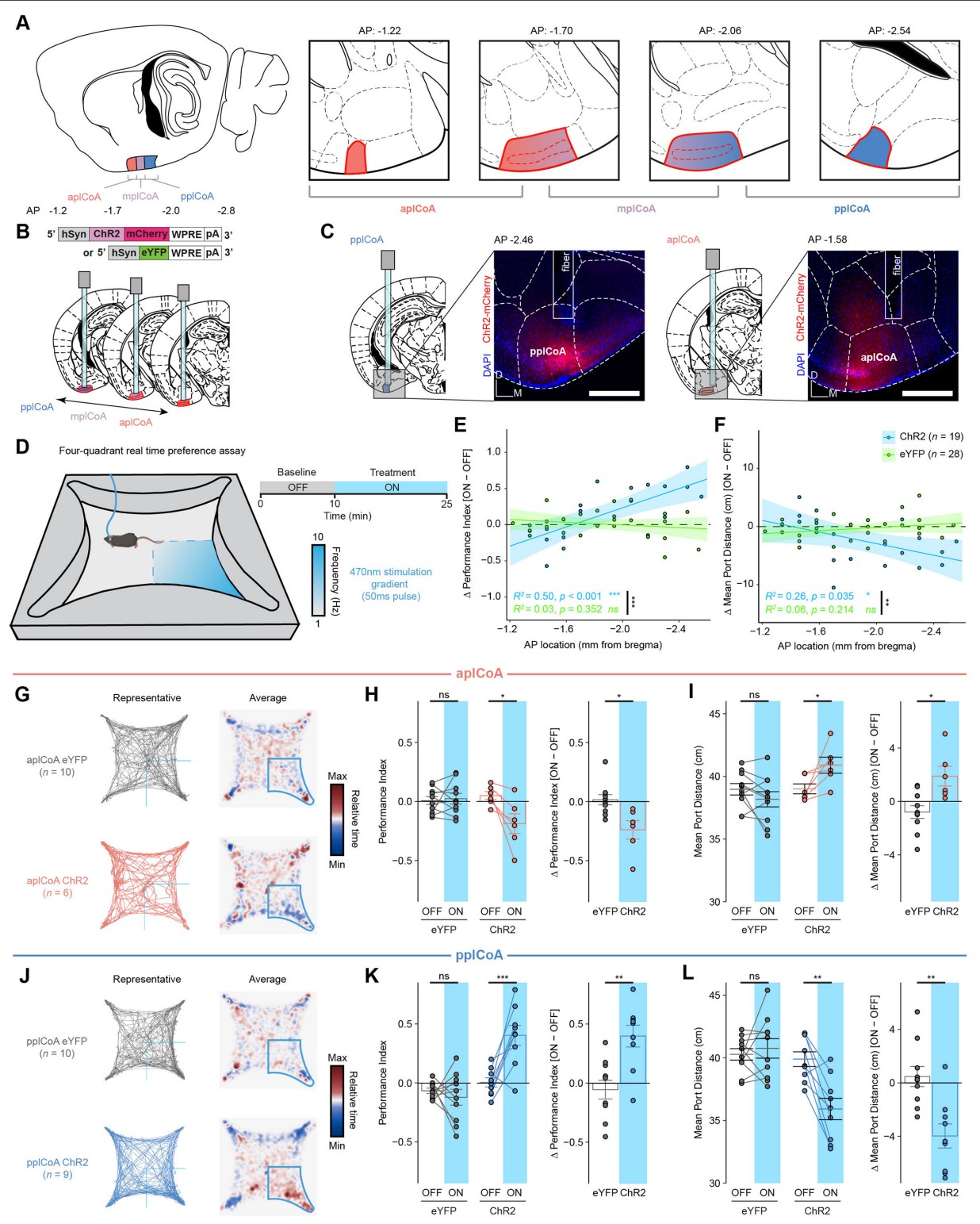

**Figure 2.** The plCoA has a topographic organization capable of driving approach and avoidance behaviors. (**A**) Schematic of plCoA domains divided into anterior (aplCoA), middle (mplCoA), and posterior (pplCoA) regions based on histology, positioning, and gradients observed in past observations (**Root et al., 2014**). (**B**) Strategy to activate anterior-posterior topographical ensembles via optogenetics. (**C**) Representative histology and fiber/virus placement for aplCoA and pplCoA ChR2 animals. Scale bar, 500 μm. (**D**) Schematic of four-quadrant open field behavioral assay with closed-loop photostimulation. (**E–F**) Linear fit of change in performance index (**E**) or mean port distance (**F**) as a function of anterior-posterior position along plCoA for optical stimulation. (**G**) Paths traveled during the stimulus period for a representative mouse (left) and baseline-normalized collective heatmaps

*Figure 2 continued on next page*

*Figure 2 continued*

(right) from both the ChR2- and eYFP-infected groups with aplCoA-localized fiber implants. Lower right stimulus quadrant indicated in blue. (**H–I**) Mean effect of Photostimulation of aplCoA neurons on time spent in stimulated quadrant (performance index) (**K**) and distance from the corner (port distance) (**I**). (**J**) Paths traveled during the treatment period for a representative mouse (left) and baseline-normalized collective heatmaps (right) from both the ChR2- and eYFP-infected groups with pplCoA-localized fiber implants. Lower right stimulus quadrant marked in blue. (**K–L**) Mean effect of photostimulation of pplCoA neurons infected with ChR2, but not eYFP, is sufficient to increase time spent in the stimulation quadrant (**K**) and reduce its average distance from the stimulation port during the stimulation period (**L**). Abbreviations: aplCoA, anterior zone of posterolateral cortical amygdala; mplCoA, middle zone of posterolateral cortical amygdala; pplCoA, posterior zone of posterolateral cortical amygdala. Across panels, ns, not significant; * p<0.05; ** p<0.01; *** p<0.001. Specific details of statistical tests can be found in *Supplementary file 1*.

The online version of this article includes the following figure supplement(s) for figure 2:

**Figure supplement 1.** Targeting of plCoA neurons for optogenetic stimulation.

**Figure supplement 2.** Activation of sparse ensembles in aplCoA and pplCoA elicits avoidance and approach.

**Figure supplement 3.** Behavioral effects of topographic plCoA stimulation are limited to valence alone.

these sparse ensembles (*Figure 2D*). Activation of aplCoA and pplCoA drove avoidance and approach responses, respectively (*Figure 2—figure supplement 2*). Notably, the magnitude of these behaviors was comparable to that observed during broad activation of each domain (*Figure 2H and K*) and did not differ significantly (*Supplementary file 1*). Although it remains possible that behavioral magnitude scales with the number of activated neurons, this question lies beyond the scope of the present study. Nonetheless, these data demonstrate that the opposing behavioral effects of anterior versus posterior plCoA activation are not artifacts of large-scale stimulation.

We further examined the effects of anterior-posterior plCoA stimulation on other behaviors to determine whether these effects were specific to appetitive and aversive responses, or if they extended to other affective or motor phenomena. Using the elevated plus maze, we found no change in anxiety based on open arm time or entries, across both the length of plCoA or within either aplCoA or pplCoA (*Figure 2—figure supplement 3A–C, E–F and H–I*). In the open field test, we similarly found no changes to thigmotaxis, based on time spent in corners of the open field, or exploration, based on time spent in the center of the open field (*Figure 2—figure supplement 3K–M, O–P and R–S*). Further, locomotion remained constant during stimulation across both assays and the entirety of plCoA (*Figure 2—figure supplement 3D, G, J, N, Q and T*). Together, these data suggest that the effects of plCoA neuron activation across the entire anterior-posterior axis are specific to approach and avoidance, with few other behavioral effects. Overall, we find that activation of plCoA neurons is sufficient to drive behaviors of opposite valence in a topographically organized manner, where aplCoA drives aversion and pplCoA drives attraction.

## Molecular diversity of transcriptomic plCoA cell types along the anteroposterior axis

Having identified a functional axis sufficient to produce approach and avoidance behaviors (*Figure 2*) that does not appear to encode valence (*Figure 1*), we next considered if the plCoA could be topographically organized by molecular cell type to support behaviors of opposing valence. Thus, we sought to determine if there is an axis of molecular cell types along the anteroposterior domains of plCoA. To investigate this phenomenon, we performed single-nucleus RNA sequencing (snRNA-seq) to determine the cell type composition and its relationship to the anterior-posterior axis of plCoA. To simultaneously profile these cell types and identify domain-specific patterns, we separately extracted tissue samples from aplCoA and pplCoA by microdissection, verified accurate dissection by histology, and pooled qualifying samples from the selected plCoA domain for each sequencing run (*Figure 3A*, *Figure 3—figure supplement 1A-D*). We also confirmed there were few region- or batch-specific differences in sequencing depth or nuclear quality markers (*Figure 3—figure supplement 1E–M*). Clustering of sequenced nuclei by gene expression allowed us to initially identify all major canonical neuronal and glial cell types in plCoA based on known marker genes identified in past scRNA-seq studies (*Figure 3B–C*; *Tasic et al., 2016*; *Zeisel et al., 2018*). Neurons in plCoA are 80% glutamatergic, while 20% of neurons are GABAergic (*Figure 3D*). We also identified large numbers of vascular leptomeningeal cells (VLMCs) and arachnoid barrier cells (ABCs), two fibroblast-like meningeal cell types that interface with vasculature and form a barrier between the brain and CSF, likely due to

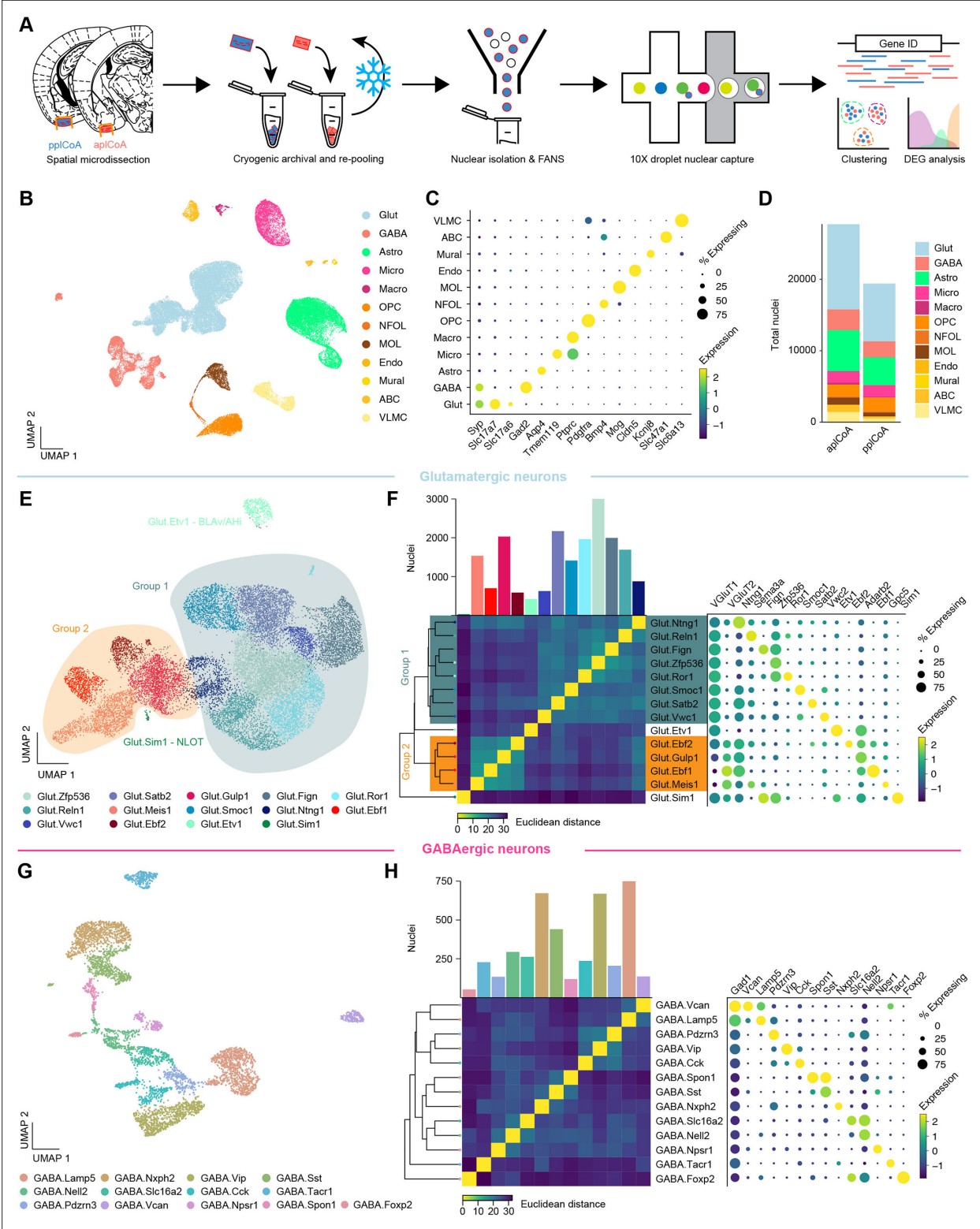

**Figure 3.** Transcriptomic heterogeneity of plCoA molecular cell types. (**A**) Schematic of freeze-and-re-pool strategy for snRNA-seq. (**B**) Two-dimensional UMAP (n=47,132 nuclei, see also **Figure 3—figure supplement 1**), colored by broad cellular identity assigned by graph-based clustering of neuronal and non-neuronal nuclei. (**C**) Cell-type-specific expression of canonical marker genes indicating broad cellular identity in the brain. Dot size is proportional to percentage of nuclei expressing the marker, with color scale representing normalized expression level. (**D**) Total proportion of cells of each identified type in each domain of plCoA. (**E**) Two-dimensional UMAP of glutamatergic neurons, colored by molecular cell type. (**F**) Clustered

*Figure 3 continued on next page*

*Figure 3 continued*

heatmap showing Euclidean distance between averages of each subtype positioned based on hierarchical clustering (left), and dot plot of marker genes for all glutamatergic subtypes (right). (**G**) Two-dimensional UMAP of GABAergic neurons, colored by molecular cell type, like in (**E**). (**H**) Clustered heatmap showing Euclidean distance between averages of each subtype positioned based on hierarchical clustering (left), and dot plot of marker genes for all GABAergic subtypes (right), like in (**F**).

The online version of this article includes the following figure supplement(s) for figure 3:

**Figure supplement 1.** Additional information and quality control for single-nucleus sequencing experiments.

**Figure supplement 2.** Additional information and quality control for spatial gene expression.

meningeal presence on the cortical surface during extraction (*Tasic et al., 2016*; *Zeisel et al., 2018*; *Marques et al., 2016*; *Yasuda et al., 2013*).

We further characterized the heterogeneity of glutamatergic and GABAergic neurons within plCoA by re-processing and subclustering both major neuronal cell types. Within glutamatergic neurons, we identified 14 distinct subtypes by gene expression, with largely continuous variation between glutamatergic subtypes (*Figure 3E*). However, when examining the relationships between these subtypes, we identified two broader groups of glutamatergic neurons via hierarchical clustering, where subtypes within each group displayed a lower Euclidean distance from one another in high-dimensional gene expression space (*Figure 3F*). Each of these two broader groups had a marker for every type within either group, where the larger Group 1 of glutamatergic neurons expresses *Slc17a7* (VGLUT1), and the smaller Group 2 expresses *Slc17a6* (VGLUT2). Within each of these glutamatergic groups, most observed marker genes are non-canonical in the amygdala and cortex, suggesting unique glutamatergic ensembles or patterns of gene expression within glutamatergic neurons in plCoA compared to other regions previously described. Two subtypes did fall outside of either broad glutamatergic group, varying in a more discrete manner than most plCoA glutamatergic neuron subtypes. Interestingly, examination of data from the Allen ISH Atlas for their respective marker genes *Etv1* and *Sim1* showed these two groups fall into adjacent regions outside of plCoA, where Glut.Etv1 neurons localize to the posterior basomedial amygdala and Glut.Sim1 neurons localize to the nucleus of the lateral olfactory tract (*Figure 3—figure supplement 2A–B*). However, gene expression patterns for GABAergic neurons displayed an opposing form of heterogeneity, where subtypes are more discrete, without broad groups linking related subtypes (*Figure 3G*). Marker genes for GABAergic neurons are also more canonical than those in glutamatergic neurons, whereby most GABAergic neurons in plCoA have interneuron-like identities, expressing canonical marker genes such as *Vip, Sst,* and *Cck* (*Figure 3H*).

We hypothesized that differences in these populations' abundance could potentially be responsible for the difference observed between different plCoA domains, and thus examined potential domain-specific enrichment of certain cell types within plCoA. Visualization of these nuclei with UMAP showed little clear region-specific structure for any major cell types (*Figure 4A*). This lack of structure was broadly confirmed quantitatively, where a few low-abundance glial cell types showed significant domain-specific enrichment, but the high-abundance major cell types did not (*Figure 4A–B*). In other brain areas, variations in the characteristics of subgroups within major cell types are more pronounced than variations in the total numbers of these major cell types, that is the balance of genes rather than balance of subtypes (*Tasic et al., 2018*; *Yao et al., 2021*). Therefore, we examined the abundance of differentially expressed genes (DEGs) between plCoA domains for each major cell type. Here, we found that both major neuronal cell types had more abundant DEGs than all major glial cell types (*Figure 4C*). Glutamatergic neurons DEGs exceeded all other major cell types by a factor of 4, suggesting that differences between the anterior and posterior domains are most likely to be observed via variation in glutamatergic neurons.

Upon examination of domain-specific variation in plCoA glutamatergic neurons, we initially observed a greater degree of domain-specific clustering in dimension-reduced space (*Figure 4D*). Glutamatergic neuron subtypes correspondingly displayed domain-specific enrichment, where more than half of glutamatergic neuron subtypes were significantly enriched in either the anterior or posterior plCoA domain (*Figure 4E*). Upon closer examination, we found that every glutamatergic subtype in the VGluT2-expressing Group 1 was enriched in anterior plCoA, while VGluT1-expressing Group 2 subtypes are evenly distributed across fields or biased towards the posterior, with one exception, Glut. Fign, which likely derives from the aplCoA-adjacent cortex-amygdala transition area CxA, based on

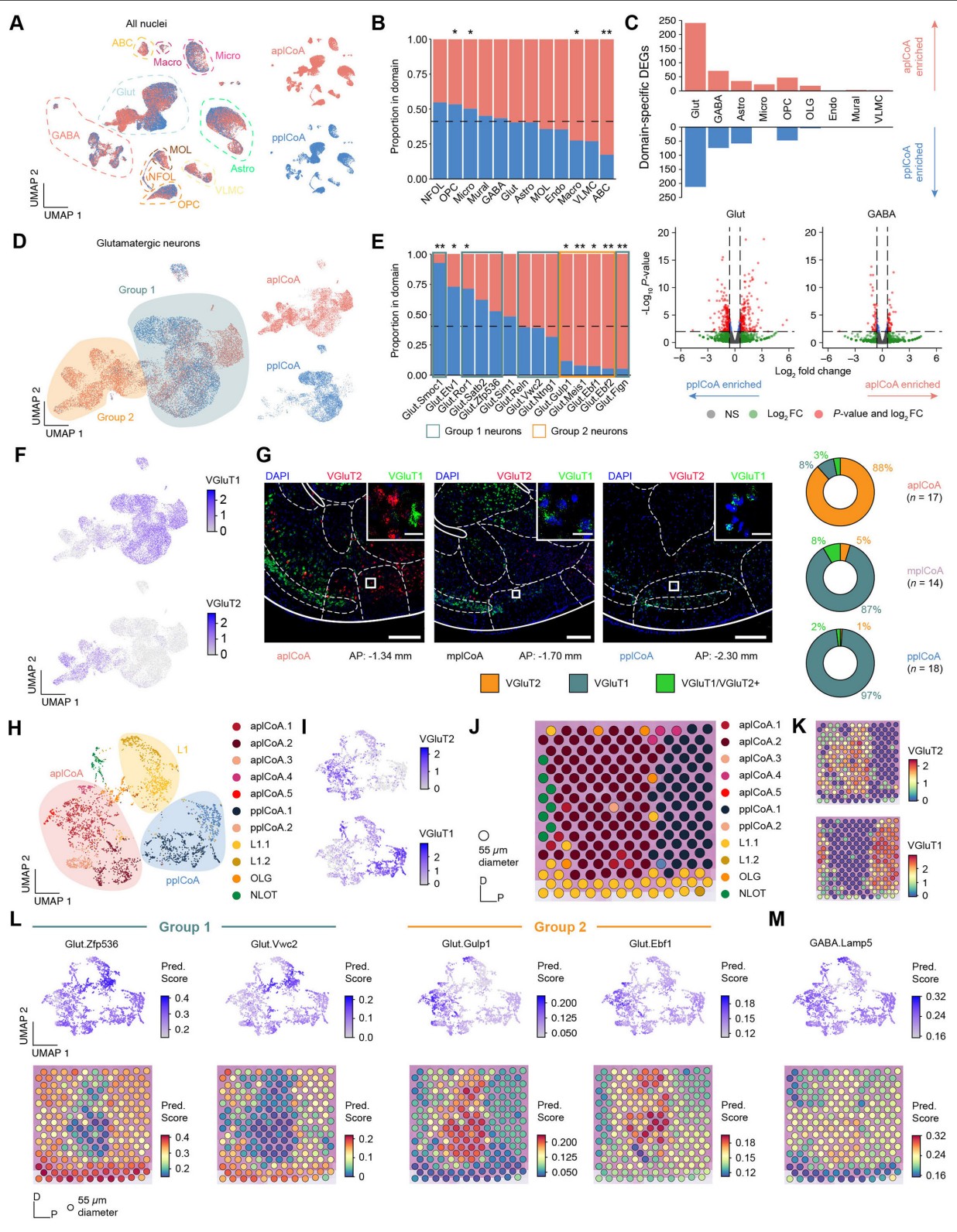

**Figure 4.** Glutamatergic neuron subtypes in plCoA are spatially distributed along an anteroposterior molecular gradient. (**A**) UMAP of all plCoA nuclei colored by zone of origin, with dotted outlines and labels denoting the major cell types. (**B**) Relative proportion of nuclei from each domain within each broad identity class. Dotted line indicates chance level for all plCoA nuclei. (**C**) Top, abundance of domain-specific DEGs for each major cell type, either enriched in aplCoA nuclei (top) or pplCoA nuclei (bottom). Bottom, volcano plots for domain-specific DEGs for glutamatergic (left) and

*Figure 4 continued on next page*

*Figure 4 continued*

GABAergic neurons (right), the two cell types with the greatest degree of domain-specific gene expression, where negative log-fold changes indicate enrichment in pplCoA and positive log-fold changes indicate enrichment in aplCoA. (**D**) UMAP of plCoA glutamatergic neurons colored by domain of origin, with dotted outlines and labels denoting the subtypes on the graph. Groups of glutamatergic neuron types identified previously via Euclidean distance and hierarchical clustering are overlaid on top of the neuron types of interest. (**E**) Relative proportion of molecular subtypes from each domain within glutamatergic neurons, where relevant subtypes are outlined according to their glutamatergic neuron group. Dotted line indicates chance level for plCoA glutamatergic neuron nuclei. (**F**) UMAP of all glutamatergic neuron nuclei, colored by expression levels of VGluT2 (top) or VGluT1 (bottom). (**G**) Left, representative images of in situ RNAscope labeling of VGluT2 RNA (red) and VGluT1 RNA (green) across plCoA domains. Right, proportions of glutamatergic neurons expressing VGluT2, VGluT1, or both. Scale bars, 500 µm (main image), 50 µm (inset). (**H**) UMAP of all plCoA-overlapping Visium capture spots, colored by cluster. Broad spatial position of groups of clusters is overlaid on top of the capture spots of interest. (**I**) UMAP of all plCoA-overlapping Visium capture spots, colored by expression levels of VGluT2 (top) or VGluT1 (bottom). (**J**) Representative plCoA-overlapping region of one section on a Visium slide capture area, with capture spots colored by cluster. (**K**) Representative plCoA-overlapping region of one section on a Visium slide capture area, with capture spots colored by expression levels of VGluT2 (top) or VGluT1 (bottom). (**L**) Prediction scores for representative glutamatergic neuron subtypes within Group 1 (left) and Group 2 (right), shown on a UMAP of all plCoA-overlapping capture spots across all sections (top) and on a representative plCoA-overlapping region of one section (bottom). (**M**) Prediction scores for a representative GABAergic neuron subtype, shown on a UMAP of all plCoA-overlapping capture spots across all sections (top) and on a representative plCoA-overlapping region of one section (bottom). Across panels: * $p<0.05$; ** $p<0.01$; *** $p<0.001$; ns, not significant. Additional specific details of statistical tests can be found in ***Supplementary file 1***.

---

Allen ISH data of *Fign* expression (***Figure 4F***, ***Figure 3—figure supplement 2A***). In contrast to glutamatergic neurons, we did not observe significant plCoA domain-specific variation for any GABAergic neuron subtypes (***Figure 3—figure supplement 2D–E***). We also found additional heterogeneity within glial cell types, including additional subtypes within astrocytes and VLMCs (***Figure 3—figure supplement 2G–J and O–R***). We also observed domain-specific divergence in gene expression for OPCs and astrocytes corresponding to domain-specific DEG differences, though they did not correspond to any observable differences in subtype enrichment (***Figure 3—figure supplement 2F and K–N***).

To confirm these findings and validate our snRNA-seq data, we directly examined spatial RNA expression patterns. First, we used RNAscope labeling to examine the expression of VGluT2 and VGluT1 in situ in the plCoA, quantifying the number of nuclei expressing these genes. We found that anterior plCoA had a much greater proportion of RNAscope-labeled VGluT2+ nuclei (88%) than VGluT1+ nuclei (11%) compared to the rest of plCoA, where VGluT1+ neurons predominate; posterior plCoA nuclei were almost entirely VGluT1+ (97%) expressing (***Figure 4G***). It is noteworthy that these numbers were generally consistent with those in our sequencing data.

The RNAscope patterns provide cellular resolution for two broadly distinct groups but lack information on the broad distribution of more nuanced subtypes along the AP axis. Thus, we next analyzed spatial gene expression in the plCoA from an existing Visium spatial transcriptomics dataset that contained sagittal sections bisecting the plCoA along the midline (***Gelber et al., 2026***; ***Figure 3—figure supplement 2S***). Although the spatial resolution is limited to 55 µm spots, this data set afforded an opportunity to test if the cell types we had identified by snRNA-seq were distributed in a spatial gradient along the plCoA axis. We asked if the domain-specific molecular cell type composition can be recovered directly from spatial information, without depending on inference from dissection histology. All sections used were of similarly high quality and did not display any clearly observable batch effects, with all but one having more than 100 spots covering the plCoA (***Figure 3—figure supplement 2T–W***). When clustering directly on spatial data, we observed significant heterogeneity separating into three broad groups (***Figure 4H***). Like in scRNA-seq, we found highly specific expression of VGluT2 and VGluT1 to two of the three broad spot groups (***Figure 4I***). When examining the spatial configuration of these groups, we found the VGluT2-expressing group of clusters was in aplCoA, while the VGluT1-expressing group was in pplCoA, with the third intermediate group corresponding to layer 1 (***Figure 4J–K***). When computationally projecting transcriptomic cell type identities onto spatial data, we observed that Group 1 glutamatergic neuron types would project onto pplCoA spots and Group 2 glutamatergic neuron types would project onto aplCoA spots, while negligible anteroposterior bias could be observed when projecting GABAergic neuron types onto plCoA spots (***Figure 4L–M***). It is important to note that this data set was collected from non-transgenic litter mate controls in a study of APP23 Alzheimer's model. The mice were of the same genetic background (C57BL/6 J) but a different age as our sequencing specimens. Thus, these data confirm the robustness of our findings from sn-RNA-seq and in situ hybridization and further demonstrate that plCoA contains

a diverse population of numerous neuronal subtypes that vary along a gradient. Whereas glutamatergic neuron subtypes vary significantly along the anteroposterior axis, such that aplCoA-enriched subtypes express VGluT2 and pplCoA-enriched subtypes express VGluT1.

## Molecularly defined plCoA glutamatergic neuron populations contribute to approach and avoidance behaviors

Given this spatial distribution bias of plCoA$^{VGluT2+}$ neurons into aplCoA and plCoA$^{VGluT1+}$ neurons into pplCoA, we further hypothesized that these glutamatergic neuron subtypes could be responsible for the opposing valence responses observed during topographic plCoA stimulation (*Figure 2*). We reasoned that if distinct molecular cell types mediate opposing valence, then activation of them in a topography-independent manner should elicit opposing responses. Therefore, we expressed ChR2 in a non-spatially-biased, cell-type-specific manner using a Cre-dependent viral construct in VGluT2-Cre and VGluT1-Cre transgenic mice, targeting AAV-DIO-hSyn-ChR2 into mplCoA and implanting optic fibers above the injection site (*Figure 5A–B*, *Figure 2—figure supplement 1B*). Using the prior four-quadrant open field task, we found that photostimulation of plCoA$^{VGluT2+}$ neurons significantly reduced time spent in the 'on' quadrant and increased the average distance to the corner port during the treatment period, indicating that activation of plCoA$^{VGluT2+}$ neurons is aversive and leads to avoidance of the quadrant when paired with stimulation (*Figure 5C–E*). In contrast, photostimulation of the plCoA$^{VGluT1+}$ neurons instead increased the time in the 'on' quadrant and decreased average distance to the corner port, indicating plCoA$^{VGluT1+}$ neuron activation is appetitive and leads to attraction to the stimulation quadrant (*Figure 5C and F–G*). These data suggest that the divergent domain-specific valence effects of plCoA activity could be due to the divergent molecularly defined neuronal ensembles predominant in each topographical field of plCoA.

Next, we sought to determine whether these two glutamatergic populations are respectively required for innate attraction and aversion to odor. We used the above transgenic mouse lines to drive expression of a viral Cre-dependent hM4D(Gi) construct to selectively inhibit these neurons' activity via chemogenetics (*Armbruster et al., 2007*; *Figure 5I*, *Figure 3—figure supplement 2*). We administered clozapine-N-oxide (CNO) or a vehicle control and used the four-quadrant open field assay (*Root et al., 2014*) to assess their behavioral responses to the innately attractive 2PE or the innately aversive TMT to determine the difference in the magnitude of temporally counterbalanced valence responses when the respective populations are chemogenetically silenced. These two odors were chosen because they most robustly drive approach and avoidance and were used in our previous work demonstrating the role of plCoA in innate responses (*Root et al., 2014*).

We observed that both transgenic mouse lines displayed attraction and aversion to 2PE and TMT following administration of the vehicle control. Inhibiting plCoA$^{VGluT2+}$ neurons by CNO administration did not affect the response to either odorant (*Figure 5J–Q*). However, inhibiting plCoA$^{VGluT1+}$ neurons abolished the attraction to 2PE, without affecting aversion to TMT (*Figure 5J–Q*). In other words, neither group of plCoA glutamatergic neurons is selectively required for TMT aversion, but plCoA$^{VGluT1+}$ neurons are required for attraction to 2PE. Further, silencing of either population did not lead to any broader non-olfactory behavior effects as measured by the EPM and OFT assays, including anxiety, exploration, and motility, showing the effects of silencing these neurons are likely limited to valence and/or olfaction alone, instead of exploratory or defensive behaviors (*Figure 5—figure supplement 1C–N*). The necessity of plCoA$^{VGluT1+}$ neurons for 2PE attraction, combined with their ability to drive approach responses with stimulation, indicates that these plCoA$^{VGluT1+}$ neurons signal attraction. However, plCoA$^{VGluT2+}$ neurons, although sufficient to drive aversion, are not necessary for aversion. Further work is required to better identify a molecularly defined population necessary for aversive responses.

## A topographic organization of plCoA defined by limbic projection targets

The anatomical connectivity of the plCoA has not yet been defined in the mouse brain. Further, we posited that the differences in necessity of plCoA cell types could be due to divergent downstream connections instead of divergent molecular features, which may partially, but not completely overlap. Thus, we next sought to identify distinct downstream outputs of plCoA that could explain the bidirectional valence effects of its topography. We first characterized the downstream outputs of

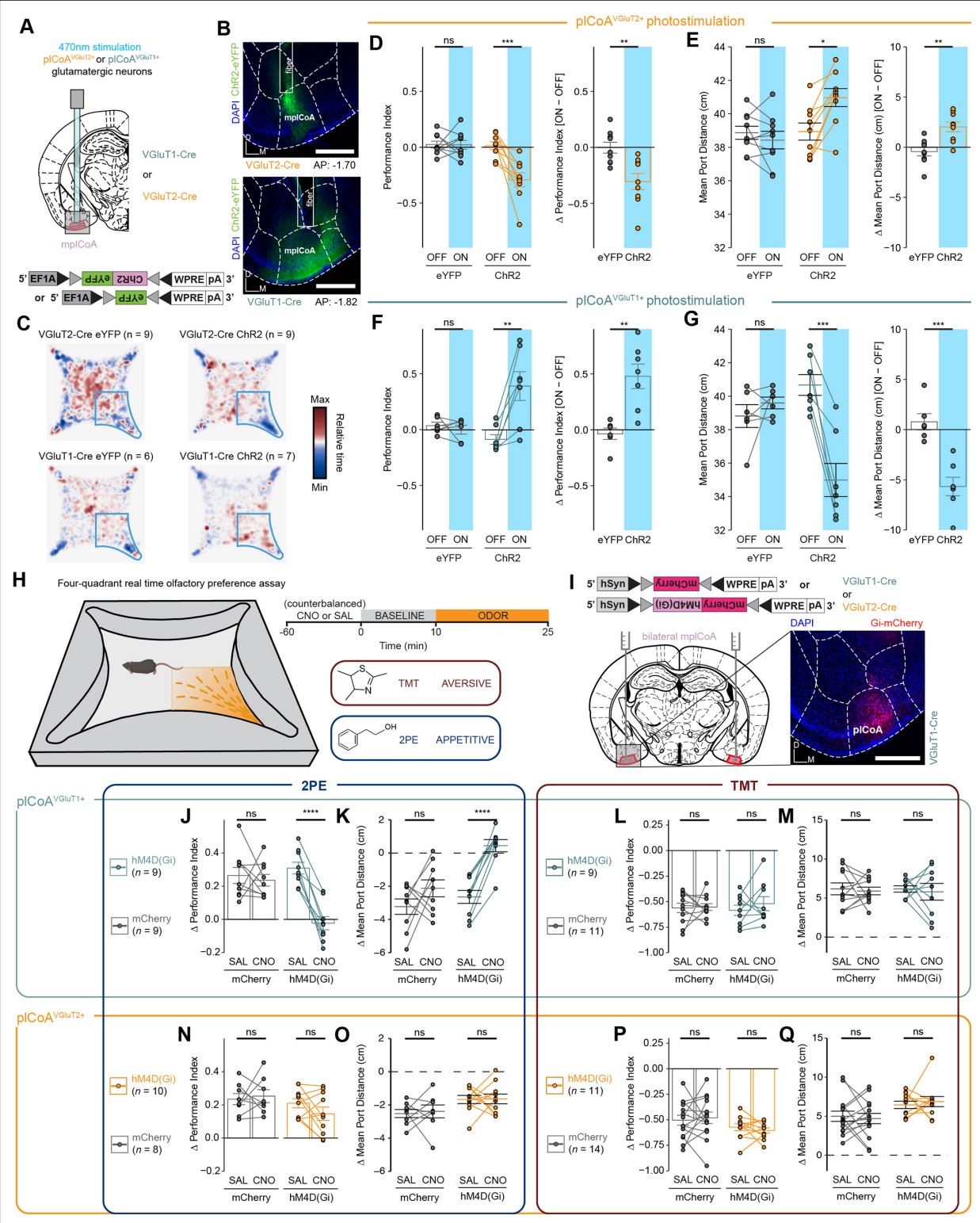

**Figure 5.** Glutamatergic plCoA[VGluT2+] and plCoA[VGluT1+] neurons contribute to innate approach and avoidance behaviors. (**A**) Schematic for selective photostimulation of distinct glutamatergic cell type. VGluT2-Cre and VGluT1-Cre animals were injected with Cre-dependent viral vectors into mplCoA with a fiber optic implant placed just above the injection site. (**B**) Representative histology from ChR2 viral injection and fiber implantation site in a VGluT2-Cre animal (top) and a VGluT1-Cre animal (bottom). Scale bar, 500 μm. (**C**) Baseline-normalized collective heatmaps from both the ChR2- and eYFP-infected groups in VGluT2-Cre and VGluT1-Cre animals with plCoA-localized fiber implants. Lower right stimulus quadrant marked in blue. (**D–G**) Effect of photostimulation of plCoA[VGluT2+] neurons (**D–E**) or plCoA[VGluT1+] neurons (**F–G**) on time spent in the stimulation quadrant (**D, F**) and distance

*Figure 5 continued on next page*

*Figure 5 continued*

from the corner (**E, G**). (**H**) Behavioral paradigm to assess innate valence responses to odor. Left, schematic of four-quadrant open field behavioral assay for spatially specific odor delivery. Upper right, within-trial timeline. Lower right, odors delivered and their associated innate valence. (**I**) Schematic of strategy for selective chemoinhibition of molecularly defined glutamatergic plCoA neurons. (**J–M**) Effect of chemoinhibition of plCoA VGluT1+ neurons on time spent in the odor quadrant (**J, L**) or decrease in mean port distance (**K, M**) in response to 2PE (**J–K**) or TMT (**L–M**). (**N–Q**) Effect of chemoinhibition of plCoA VGluT2+ neurons on time spent in the odor quadrant (**N, P**) or decrease in mean port distance (**O, Q**) in response to 2PE (**N–O**) or TMT (**P–Q**). Across panels, ns, not significant; * p < 0.05; ** p < 0.01; *** p < 0.001, **** p < 0.0001. Additional specific details of statistical tests can be found in **Supplementary file 1**.

The online version of this article includes the following figure supplement(s) for figure 5:

**Figure supplement 1.** Additional information for Cre-dependent molecularly targeted chemogenetic inhibition experiments.

plCoA by co-injecting the anterograde viral tracer AAV-DIO-hSyn-mRuby2-T2A-synaptophysin-EGFP and a constitutive AAV-Cre virus into mplCoA to label presynaptic terminals with EGFP (***Figure 6—figure supplement 1A–B***). We observed a high amount of terminal fluorescence within plCoA itself, suggesting the presence of recurrent connections within the region (***Figure 6—figure supplement 1B***). In addition, we observed long-range projections to a diverse set of regions, including surrounding extended amygdala subregions, such as MeA and the amygdalo-hippocampal transition area (AHi), regions controlling valence and emotion, like the NAc and the bed nucleus of the stria terminalis (BNST), and regions involved in olfactory processing, primarily PIR and OT (***Figure 6A–D***). The projection to the NAc is broad without any apparent spatial restriction, innervating both the core and shell.

Among these outputs, we hypothesized that the NAc and the MeA could be responsible for the behavioral divergence between anterior and posterior plCoA, given their known involvement in reward expectation and aversion, respectively (***Müller and Fendt, 2006***; ***Ikemoto, 2007***). We sought to confirm these differences using retrograde tracing from MeA or NAc, where red retrobeads were injected into MeA or NAc and labeled neurons were quantified along the anterior-posterior axis (***Figure 6E***). For both downstream targets, we observed opposing gradients of retrobead projector labeling throughout the entire plCoA anteroposterior axis (***Figure 6F***). MeA-projecting neurons are enriched in aplCoA, and NAc-projecting neurons are enriched in pplCoA, with each having a frequency of around chance level in mplCoA (***Figure 6G***). Further, the majority of labeled plCoA-MeA projection neurons were in aplCoA, while the majority of labeled plCoA-NAc projection neurons were in pplCoA (***Figure 6H***). To further confirm the spatial bias in projection targets, we performed anterograde tracing from the aplCoA and pplCoA by injecting viruses expressing either eYFP or mCherry into either domain of plCoA in a counterbalanced manner (***Figure 6—figure supplement 1D–E***). Anterograde projection strength from aplCoA and pplCoA revealed that projections to MeA were most dense from aplCoA, and projections to NAc were most dense from pplCoA (***Figure 6I–K***). The aplCoA sent a significantly higher proportion of its projections to MeA than pplCoA, whereas pplCoA sent a significantly higher proportion of its projections to NAc (***Figure 6L***).

We reasoned that the topographical sufficiency we observed (***Figure 2***) could be explained by cell-type-specific divergence in projection target, whereby the topographical biases in downstream targets are recapitulated by their underlying molecular cell type. To determine the relationship between cell types and projection targets, we injected Cre-dependent eYFP into mplCoA in VGluT2-Cre and VGluT1-Cre transgenic mice, targeting the middle to ensure differences result from cell type, instead of simply redundant topography (***Figure 6—figure supplement 1F–G***). Interestingly, the relationship was not as simple as one cell type, one primary projection target. Rather, both cell types project to both structures in different proportions. The plCoA$^{VGluT2+}$ neurons primarily project to MeA with a significant bias for that target over NAc, while plCoA$^{VGluT1+}$ neurons project to both MeA and NAc, with a statistically insignificant bias toward NAc (***Figure 6M and P***). These findings demonstrate that NAc primarily receives projections from plCoA$^{VGluT1+}$ neurons, whereas the MeA receives input from both populations. Moreover, projection of both cell types to the MeA may explain why neither VGluT population was selectively required for innate aversion (***Figure 5J–Q***).

We further characterized VGluT expression in projection-defined neurons by combining retrograde tracing from either the MeA or NAc with in situ RNAscope for VGluT1 and VGluT2 (***Figure 6—figure supplement 2***). Given that the anterior and posterior regions of plCoA are predominately VGluT2 and VGluT1 positive (***Figure 4G***), respectively, and both regions send projections to the MeA and NAc (***Figure 6M–P***), we focused our analysis on the middle plCoA, which contains a more balanced mixture

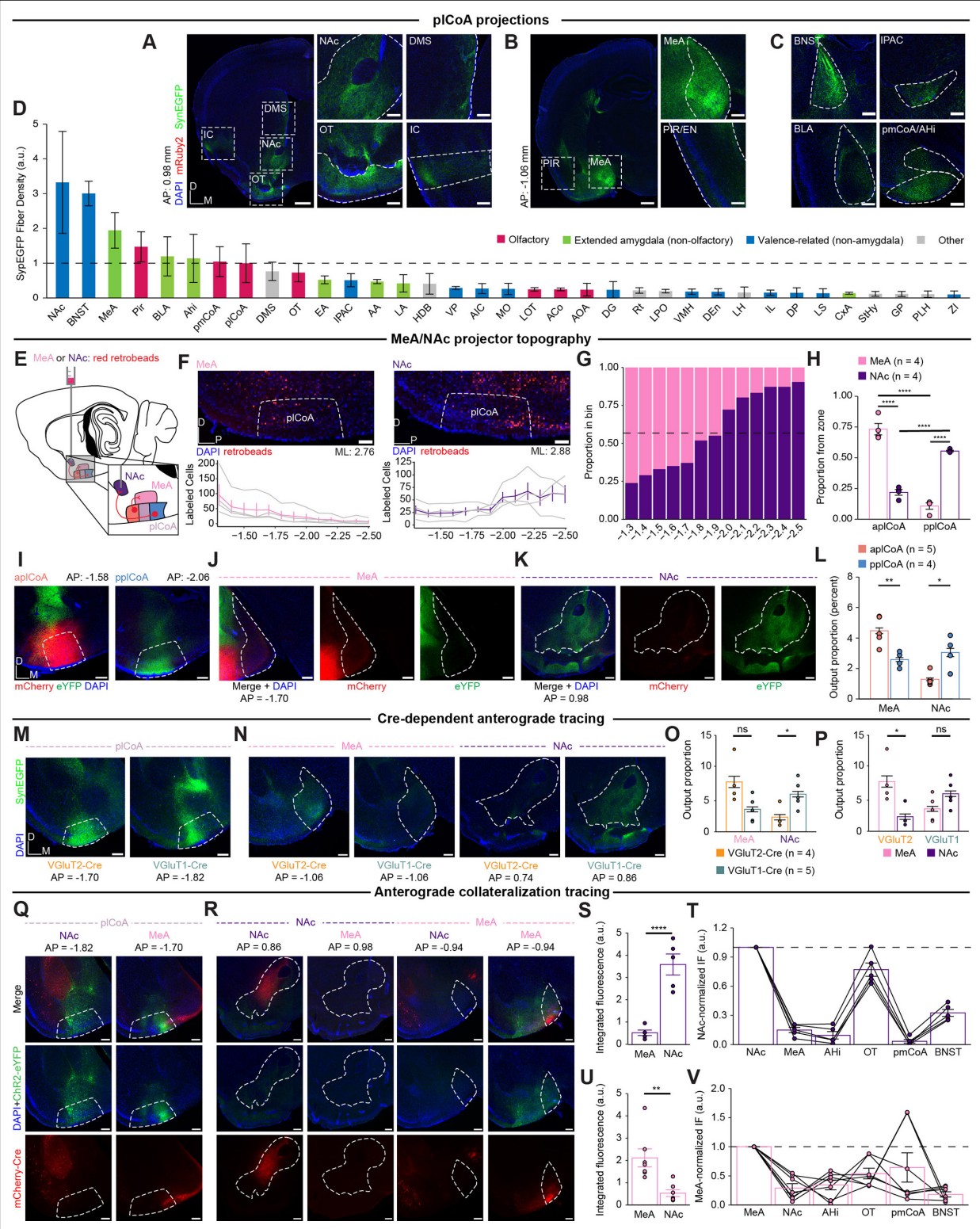

**Figure 6.** Projections to MeA and NAc from plCoA are topographically organized. (**A**) Left, whole-hemisphere view at AP = 0.98 mm from bregma. Scale bar, 500 μm. Right, Magnified images of the areas highlighted inside white dashed lines. Scale bar, 200 μm. (**B**) Left, whole-hemisphere view at AP = –1.06 mm from bregma. Scale bar, 500 μm. Right, Magnified images of the areas highlighted inside white dashed lines. Scale bar, 200 μm. (**C**) Other plCoA projections not found in cross-sections of the brain found in (**A**) and (**B**). Scale bar, 200 μm. (**D**) Magnitude of anterograde synaptophysin-eYFP fluorescence in primary downstream targets of plCoA projection neurons ordered by total output strength, colored based on each region's function. (**E**) Schematic for topographic retrograde mapping strategy from MeA and NAc into plCoA. Red retrobeads are injected into MeA or NAc and

*Figure 6 continued on next page*

*Figure 6 continued*

topographical projection bias is examined along the anterior-posterior axis. (**F**) Representative images (top) for injection into MeA (left) or NAc (right) and number of neurons labeled along the anterior-posterior axis as distance (mm) from bregma (bottom). Gray lines denote individual replicates, where colored lines indicate mean ± s.e.m. (**G**) Proportion of retrobead-labeled neurons projecting to MeA or NAc for each 100 μm segment as a function of distance from bregma. Dashed line indicates overall balance of all retrobead-labeled neurons across entire plCoA. (**H**) Proportion of retrobead-labeled neurons from either target within each plCoA zone. MeA-labeled neurons are significantly enriched in aplCoA compared to NAc-labeled neurons, while NAc-labeled neurons are significantly enriched in pplCoA compared to those labeled from MeA. (**I**) Representative histological images for the injection sites in aplCoA (left) and pplCoA (right) from a representative animal. Scale bar, 200 μm. (**J**) Representative histological images for MeA from the animal in (**J**). Scale bar, 200 μm. (**K**). Representative histological images for NAc from the animal in (**J**). Scale bar, 200 μm. (**L**) Output strength as a proportion of total fluorescence from aplCoA and pplCoA to MeA and NAc. (**M**) Representative histological images for the injection site in plCoA from a representative VGluT1-Cre and VGluT2-Cre animal. Scale bar, 200 μm. (**N**) Representative histological images from MeA and NAc from a representative animal of either genotype. Scale bar, 200 μm. (**O**) Left, output strength as a proportion of total fluorescence from plCoA$^{VGluT2+}$ and plCoA$^{VGluT1+}$ neurons to MeA and NAc. Right, comparison of same data, but by target region within genotype. (**P**) Same data as (**O**), but by target region within genotype. (**Q–T**) Mapping collateral projections from NAc- and MeA-projecting neurons. (**Q**) Representative histological images for the injection site in plCoA from a representative animal receiving retrograde virus into MeA or NAc. Scale bar, 200 μm. (**R**) Representative histological images of NAc and MeA retro-Cre targeting (red) and outputs (green). (**S**) Comparison of absolute integrated fluorescence intensities in MeA and NAc when retroAAV was injected into NAc (top) or MeA (bottom). (**T**) Quantification of fluorescence in selected downstream brain regions from plCoA originating from plCoA-NAc neurons proportional to eYFP fluorescence in NAc (top) or MeA (bottom). Abbreviations: NAc, nucleus accumbens; BNST, bed nucleus of stria terminalis; MeA, medial amygdala; Pir, piriform cortex; BLA, basolateral amygdala; Ahi, amygdalo-hippocampal transition area; pmCoA, posteromedial cortical amygdala; Str, striatum; OT, olfactory tubercle; EA, extended amygdala; IPAC, inferior peduncle of the anterior commissure; AA, anterior amygdala; LA, lateral amygdala; HDB, horizontal limb of the diagonal band; VP, ventral pallidum; AIC, anterior insular cortex; mfb, medial forebrain bundle; MO, medial orbitofrontal cortex; LOT, lateral olfactory tract; ACo, anterior cortical amygdala; AOA, anterior olfactory area; DG, dentate gyrus; Rt, reticular nucleus; LPO, lateral preoptic area; VMH, ventromedial hypothalamus; DEn, dorsal endopiriform claustrum; LH, lateral hypothalamus; IL, infralimbic cortex; DP, dorsal peduncular cortex; LS, lateral septum; CxA, cortex-amygdala transition area; sox, supraoptic decussation; StHy, striohypothalamic nucleus; GP, globus pallidus; PLH, perirhinal cortex; ZI, zona incerta. Across panels, ns, not significant; * p<0.05; ** p<0.01; **** p<0.0001. Additional specific details of statistical tests can be found in ***Supplementary file 1***.

The online version of this article includes the following figure supplement(s) for figure 6:

**Figure supplement 1.** Additional information for plCoA anterograde tracing experiments.

**Figure supplement 2.** Analysis of VGluT expression in MeA- and NAc-projecting neurons.

of these cell types. In this region, we found that most MeA-projecting neurons (62%) expressed VGluT2, whereas only a small fraction (4%) expressed VGluT1. In contrast, NAc-projecting neurons were largely VGluT1-positive (64%), with an additional 20% co-expressing both VGluT1 and VGluT2. A subset of retrogradely labeled neurons appeared negative for both markers, which may reflect detection limits or the presence of non-glutamatergic projection neurons. Overall, these results align with the anterograde tracing data, indicating that VGluT2-expressing neurons predominantly project to the MeA, VGluT1-expressing neurons primarily project to the NAc, and a smaller population projects to both targets.

Given that both cell types project to both MeA and NAc, we sought to determine the extent of collateralization in neurons composing the two pathways. To test whether plCoA-MeA and plCoA-NAc projection neurons also project to multiple or overlapping downstream targets, we employed a combination of retrograde Cre and Cre-dependent anterograde tracer viral vectors. A retroAAV-hSyn-Cre-mCherry virus was injected into either MeA or NAc, and AAV-DIO-ChR2-eYFP was injected into plCoA to label outputs of MeA- or NAc-projecting neurons (***Figure 6Q–R***, ***Figure 6—figure supplement 1H–I***). We focused on MeA and NAc, as well as the ancillary primary downstream targets implicated in valence or olfaction. We found different collateralization patterns for both populations, where NAc-projecting neurons did not collateralize to MeA, but very strongly collateralized to OT. In contrast, MeA-projecting neurons minimally collateralized to NAc and most strongly collateralized to pmCoA (***Figure 6S-T***). Notably, neither projection of interest significantly collateralized to the other. These data indicate that plCoA-MeA and plCoA-NAc projection neurons are largely non-overlapping, spatially biased populations that output to different downstream subnetworks.

## The plCoA neurons projecting to the NAc and MeA respectively mediate attraction and aversion to odor

The topographic separation of MeA- and NAc-projecting neurons is consistent with a model of divergence valence that could support the observed topographic divergence behaviors. To investigate the

behavioral contributions of these projections, we first determined whether the neurons projecting to the MeA and NAc are able to drive behavior with optogenetic stimulation. We expressed ChR2 in a non-spatially biased manner by injecting AAV-hSyn-ChR2 into mplCoA, and we placed an optic fiber above MeA or NAc for selective optogenetic stimulation at plCoA axon terminals (*Figure 2—figure supplement 1C*). We found that photostimulation of the plCoA-MeA circuit in the four-quadrant open field task significantly reduced time spent in the 'on' quadrant and increased the average distance to the corner port during the treatment period, indicating activation of the plCoA-MeA circuit is aversive and leads to avoidance of the quadrant paired with stimulation (*Figure 7A–D*). The opposite was true for the plCoA-NAc projection, where stimulation in that zone instead increased the time in the 'on' quadrant and decreased average distance to the corner port, indicating activation of the plCoA-NAc circuit is instead appetitive and leads to attraction to the stimulation quadrant (*Figure 7B and E–F*). We next asked if the effects of stimulating these circuits affected other non-valence behaviors by testing the mice in the EPM and OFT. Using the EPM, we found no change in anxiety based on open arm time or entries when stimulating either projection to the MeA or NAc (*Figure 7—figure supplement 1A–C, E–F and H–I*). Similarly, stimulation in the OFT did not cause any change to thigmotaxis, based on time spent in corners of the open field, or exploration, based on time spent in the center of the open field (*Figure 7—figure supplement 1K–M, O–P and R–S*). Further, locomotion remained constant during stimulation across both assays (*Figure 7—figure supplement 1D, G, J, N, Q and T*). These data indicate that the divergent projections from plCoA to the MeA and the NAc are capable of driving valence-specific behaviors without modulating anxiety.

Finally, we sought to determine whether plCoA-MeA or plCoA-NAc projecting neurons are necessary for the expression of odor-evoked appetitive or aversive behaviors. To target these projection neurons for chemogenetic silencing, we injected a retroAAV bearing an hSyn-EBFP-Cre construct into MeA or NAc, along with an AAV in plCoA bearing a Cre-dependent hM4D(Gi) construct (*Figure 7G and H*). This chemogenetic strategy was chosen over optogenetic inhibition based on suitability for minutes-long inhibition and ease of movement in a closed arena. We then tested the innate responses of these animals to 2PE or TMT in the four-quadrant assay following administration of CNO or a vehicle control, as above. Inhibition of plCoA-NAc projecting neurons abolished innate attraction to 2PE without having any effect on aversion to TMT (*Figure 7I–L*). Conversely, inhibition of plCoA-MeA projecting neurons had no effect on innate 2PE attraction, but significantly decreased the aversion to TMT (*Figure 7M–P*). Silencing these neurons had no effect in the EPM and OFT assays, indicating the effects of silencing these neurons are limited to valence or olfaction, and not anxiety or exploration (*Figure 7—figure supplement 1M–X*). Thus, plCoA-MeA projecting neurons are necessary and sufficient for innate aversion to TMT, whereas plCoA-NAc projecting neurons are necessary and sufficient for innate attraction to 2PE.

## Discussion

### Topographic organization of valence circuitry in plCoA

The neural circuits mediating innate attraction and aversion to odor have not been fully defined. Here, we have advanced our knowledge of the circuitry underlying innate olfactory behaviors by defining its activity and organization within plCoA and further extending the innate olfactory pathway from a third-order olfactory brain area to limbic structures involved in motivational valence. We have identified a novel functional axis for valence with the plCoA that is defined by histologically and functionally distinct domains along the anteroposterior axis. We have characterized odor encoding in plCoA, identifying a sparse population code for identity of an odor, but not its innate valence, consistent with previous findings (*Iurilli and Datta, 2017*). We have also determined the composition of molecular cell types in plCoA and identified spatially biased populations enriched within each domain, which we find sufficient to drive their respective domain-specific behaviors, though only partially necessary for their functions in olfaction. Moreover, we identified the outputs of plCoA and quantitatively characterized the relative anatomical strength of each, as well as how it relates to plCoA topography and domain-specific molecular cell types, demonstrating that neurons projecting to the MeA and NAc are topographically and molecularly biased. Finally, we demonstrate that neurons projecting to the to the NAc and MeA are capable of driving approach and avoidance responses, and loss of function experiments demonstrate that the neurons projecting to the NAc or MeA selectively support innate olfactory

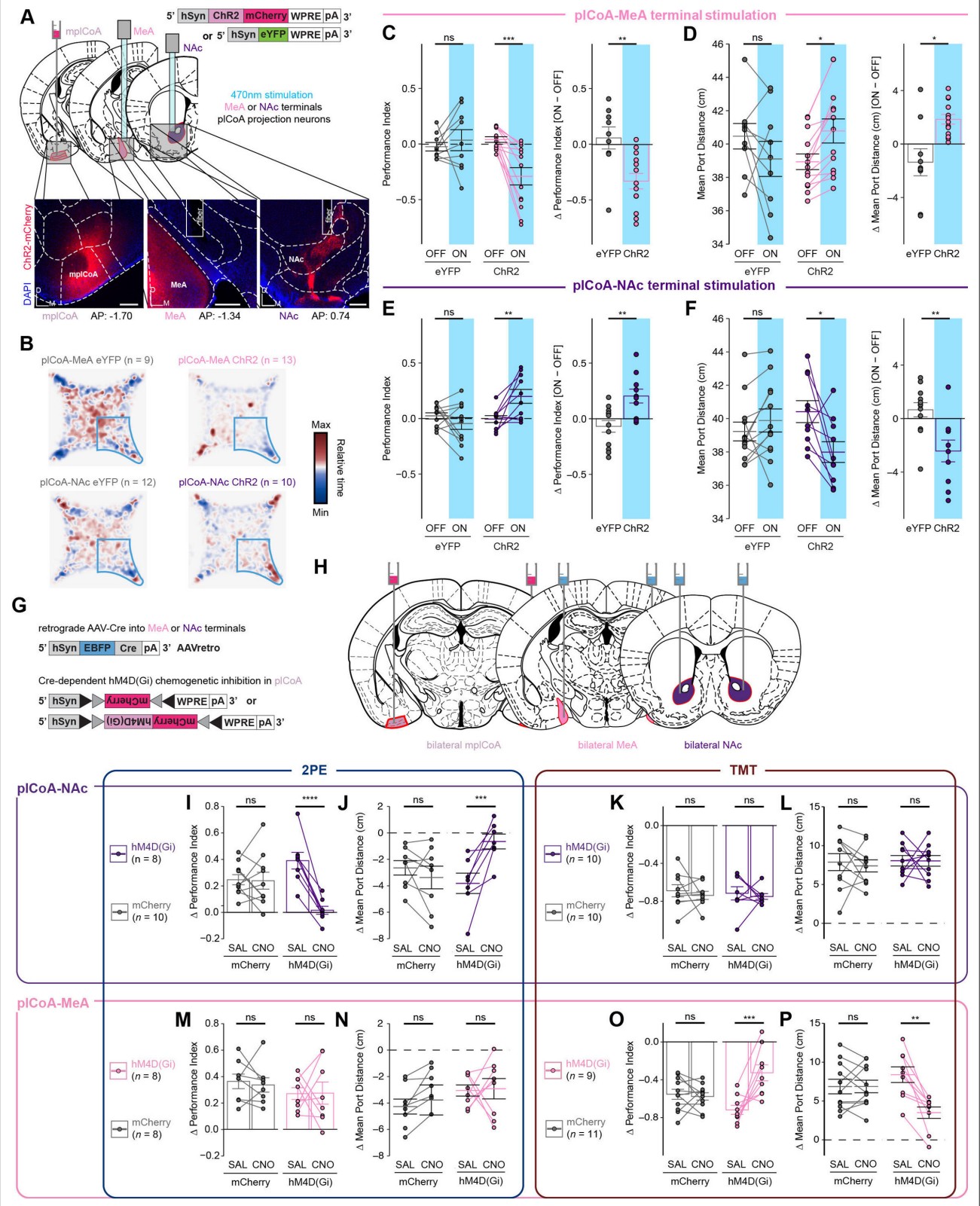

**Figure 7.** Divergent neurons projecting from plCoA to NAc and MeA control innate olfactory attraction and aversion. (**A**) Schematic for optogenetic MeA terminal stimulation in plCoA neurons. Strategy to activate MeA-projecting plCoA neuron terminals via optogenetics (top) and representative histology from ChR2 viral injection and fiber implantation site (bottom). Scale bars, 200 μm. (**B**) Baseline-normalized collective heatmaps from both the ChR2- and eYFP-infected plCoA groups with MeA- and NAc-localized fiber implants. Lower right stimulus quadrant marked in blue. (**C–D**) Optogenetic

*Figure 7 continued on next page*

*Figure 7 continued*

MeA terminal stimulation of plCoA neurons infected with ChR2, but not eYFP, is sufficient to reduce time spent in the stimulation quadrant (**C**) and increase its average distance from the port (**D**) during the stimulation period. (**E–F**) Optogenetic NAc terminal stimulation of plCoA neurons infected with ChR2, but not eYFP, is sufficient to increase time spent in the stimulation quadrant (**E**) and decrease its average distance. from the stimulation port (**F**) during the stimulation period. (**G**) Viral strategy for selective retrograde chemoinhibition of projection-defined plCoA neurons. (**H**) Schematic for selective retrograde chemoinhibition of projection-defined plCoA neurons. (**I–L**) Chemoinhibition of NAc-projecting plCoA neurons significantly eliminates the 2PE-evoked increase in time spent in the odor quadrant (**I**) and decreases in mean port distance (**J**). The response to TMT is unaffected in time spent in odor quadrant (**K**) or port distance (**L**). (**M–P**) Chemoinhibition of MeA-projecting plCoA neurons does not affect 2PE-evoked increase in time spent in the odor quadrant (**M**) or decrease in mean port distance (**N**) significantly decreases the TMT-evoked reduction in time spent in the odor quadrant (**O**) or increase in mean port distance (**P**). Across panels, ns, not significant; * p<0.05; ** p<0.01; *** p<0.001. Additional specific details of statistical tests can be found in *Supplementary file 1*.

The online version of this article includes the following figure supplement(s) for figure 7:

**Figure supplement 1.** Manipulation of plCoA projections to MeA or NAc in either direction does not change features of behavior unrelated to innate valence.

attraction and avoidance, respectively. Thus, plCoA is composed of dissociable, spatially segregated ensembles for divergent valence, defined by their downstream projection target. Though, it remains to be determined how broadly these neurons contribute to the valence of other odors.

Spatial organization for features of sensory stimuli is common in sensory cortex and has been previously proposed as fundamental to sensory processing (*Kaas, 1997*). Visual cortex is topographically organized by retinotopic locations in space, somatosensory cortex contains a map of the body, auditory cortex has a crude tonotopic organization by frequency, and gustatory cortex is segregated by taste qualities (*Chen et al., 2011*; *Garrett et al., 2014*; *Kaas et al., 1979*; *Schreiner and Winer, 2007*). However, topographic organization by perceptual feature has not been observed in the olfactory system, which has been attributed to the high dimensional nature of olfactory information (*Stettler and Axel, 2009*). Our findings identify a topographically organized divergence motif for valence in plCoA, where activation of aplCoA evokes aversive behavioral responses, and activation of pplCoA evokes attractive behavioral responses, with a graded transition between the two domains (*Figure 2D–F*). This is consistent with prior studies implicating a spatial organization to inputs from the OB to plCoA, where glomerulus-specific anterograde tracing from OB shows specific glomeruli send fibers to invariant, densely clustered, anatomically distinct locations within plCoA, and retrograde tracing from plCoA shows that upstream glomeruli are spatially biased within OB (*Miyamichi et al., 2011*) and closer examination reveals that OB input can co-vary with plCoA anteroposterior position. Interestingly, spatial segregation of divergent features is present elsewhere in olfaction as well, although not by perceptual feature. Olfactory sensory receptors display stereotyped spatial organization within zones of the olfactory epithelium, with a corresponding spatially-stereotyped glomerular topography in OB (*Soucy et al., 2009*; *Wang et al., 2022*; *Ressler et al., 1993*; *Zapiec and Mombaerts, 2020*). Further, these topographic domains in OB have functional relevance, where region-specific OB manipulations selectively alter different olfactory behaviors (*Kobayakawa et al., 2007*; *Kermen et al., 2016*). However, the piriform cortex generally lacks apparent spatial organization or spatial patterning for odor responses (*Stettler and Axel, 2009*). Thus, the topographical organization of valence observed here in plCoA represents one of the first descriptions of these spatial patterns occurring in central olfactory areas.

Topographic gradients for appetitive and aversive responses have been observed in other limbic regions, including the BLA (*Kim et al., 2016*), the MeA for innate social behaviors (*Choi et al., 2005*), *Drosophila* dopaminergic mushroom body neurons for olfactory learning (*Cohn et al., 2015*), and the gustatory insular cortex for taste (*Chen et al., 2021*). Our results extend this phenomenon of spatially segregated neurons for valence into a less well-described amygdala nucleus and suggest that it could serve as a potential common motif within the limbic system to organize motivational information, especially for innate behaviors, which require stereotyped neurocircuitry. We believe a model whereby aplCoA and pplCoA are parts of the same region with similar underlying composition, but with a gradual change in the factor that defines the valence output for a given part of the gradient would best explain the underlying gradient-like effect on approach and avoidance. We considered two such organizational principles underlying this topography: molecular cell types and projection targets. We found that broad divisions of cell types by VGluT2 and VGluT1 did not fully define valence, though

more specific subtypes could be responsible. Our results indicate that the projection targets of plCoA neurons are a fundamental feature for imparting valence on this circuitry.

## Molecular cell types in plCoA segregate topographically and support behaviors of divergent valence

While examining the cell types composing plCoA, we noticed numerous notable, novel features. First, despite its small area (~1.04 mm$^3$ and ~170,000 cells *Erö et al., 2018*), plCoA displays remarkable diversity, hosting dozens of distinct, robustly separable cell types (*Erö et al., 2018*). The plCoA appears to have multiple domains positioned at the transition between disparate brain tissue types. Interestingly, the high-dimensional structure of molecular variation differs between glutamatergic and GABAergic neurons. Glutamatergic molecular variation within plCoA is continuous, with two broadly nested groups generally marked by either VGluT2 or VGluT1 along with one or more additional marker gene(s), though it should be noted that expression of the two broad glutamatergic markers is not necessarily mutually exclusive and a few low abundance 'transition' cell types can express both (*Figure 3E–F*). This leaves open the possibility that a more specific cell type could be necessary for aversion, given that this necessity for this behavior did not map onto either broad molecular cell type. In contrast, molecular variation in GABAergic neurons is far more discretized, with ensembles expressing one of a few well-characterized interneuron markers found throughout the brain, such as *Sst, Vip,* and *Pvalb*, among others (*Figure 3G–H*). This is consistent with other studies in neocortex, hippocampus, and subiculum that find similar patterns of variation, whereby variation within glutamatergic neurons is more continuous than in GABAergic neurons (*Tasic et al., 2018*; *Cembrowski et al., 2018*). In this manner, we find the continuous gradient-like structure of valence in plCoA is recapitulated with gradient-like variation in glutamatergic neuron gene expression in plCoA.

In these molecular datasets, we further observed specific differences in cell type enrichment within aplCoA and pplCoA within glutamatergic neurons, but not GABAergic neurons or glia. Within plCoA, VGluT2+ neurons are enriched in aplCoA and VGluT1+ neurons are enriched in pplCoA, although there is gradient-like intermingling of populations, especially toward the middle of plCoA, and all glutamatergic neuron types are present, albeit with high variability along the anteroposterior axis. This molecularly defined order suggests a programmed organization, rather than stochastically distributed populations within the region, especially given that its boundaries match the domains previously identified based on behavior and histology (*Cembrowski and Spruston, 2019*). This phenomenon also broadly matches observations in the neocortex, hippocampus, and subiculum, where glutamatergic neurons across subdivisions molecularly diverge to a greater degree than GABAergic neurons or glia, albeit across a correspondingly greater area than within plCoA, which is generally accompanied by distinct morphological and electrophysiological properties broadly corresponding to these transcriptomic differences (*Tasic et al., 2018*; *Yao et al., 2021*; *Cembrowski et al., 2018*; *Scala et al., 2021*). Spatial segregation of molecular cell types is also observed within deeper brain regions including BLA, thalamus, and habenula, and these molecular differences are also accompanied by extended phenotypic differences (*Kim et al., 2016*; *Calvigioni et al., 2023*; *Phillips et al., 2019*; *Mandelbaum et al., 2019*). Investigation of such properties held in common and diverging within and between VGluT2+ and VGluT1+ glutamatergic neuron types could also serve to further define the local neurocircuitry and information processing dynamics within plCoA and along its anteroposterior axis.

Interestingly, few if any populations within plCoA are clearly separable from most of the regions surrounding it (e.g. piriform cortex, basomedial amygdala, and MeA) based on primary marker gene identity. Instead, plCoA ensembles seem to be defined by the interplay of all three regions within the same tissue. The predominance of populations resembling different regions does appear related to this anteroposterior organization, though, where VGluT2+ neurons predominate in both aplCoA and MeA, and more specific marker genes like *Meis2* are expressed in both regions as well (*Hochgerner et al., 2023*). Conversely, VGluT1+ neurons predominate in both pplCoA and piriform cortex, and the major marker genes like *Satb2* are similarly expressed in both regions (*Klingler, 2017*). Such phenomena are also consistent with general characterizations made in whole-brain molecular taxonomies, which divide plCoA along its axis, grouping aplCoA with MeA and pplCoA with paleocortex (*Shi et al., 2023*; *Yao et al., 2023*; *Zhang et al., 2023*). It would be misleading to characterize plCoA populations as mere extensions of surrounding populations into an adjacent region, though. VGluT1+ neurons from the cortex-amygdala transition zone are also present in the dataset and are continuously

separable from VGluT1+ plCoA neurons based on the expression of marker genes like *Fign*. Instead, plCoA may itself be a transition region, given that such a relationship with its neighboring regions is very similar to that of the amygdalostriatal transition area, one of the only transition regions to undergo high-resolution molecular profiling (*Mills et al., 2022*). Given such commonalities between these two putatively dissimilar regions, molecular characterization of additional transition regions could potentially uncover similar organizational motifs, especially if compared with adjacent regions of interest, and allow for a much more in-depth exploration and characterization of the boundaries and transitions between proximally located, distantly related brain regions.

These broad molecular groups of glutamatergic cell types themselves do not completely explain valence in the plCoA. While aplCoA-enriched VGluT2+ neurons are sufficient to drive aversion and pplCoA-enriched VGluT1+ neurons are sufficient to drive attraction (*Figure 7A–F*), as would be predicted from the valence responses evoked from each anterior-posterior domain enrichments, it might be expected that these populations would also be necessary for the respective odor-evoked valence. However, only VGluT1+ neurons are necessary for 2PE attraction, whereas the VGluT2+ neurons were not required for aversion to TMT (*Figure 7I–P*). Given that the plCoA is necessary for TMT aversion, it is unlikely that such a difference is due to additional redundant function within other regions for TMT aversion (*Root et al., 2014*). Rather, although VGluT2+ neurons likely contribute to aversion, other populations within the region not expressing the marker gene could also contribute to the behavioral response. Thus, although these two broad glutamatergic groups can drive innate responses of valence, the VGluT2+ population doesn't fully represent the population that controls aversive responses, which is supported by anatomic tracing presented here showing that both VGluT2+ and VGluT1+ neurons project to MeA and NAc to different degrees (*Figure 6M–P*, *Figure 6—figure supplement 2*). Regarding more specific molecular cell types, it could be possible that only a subset of VGluT1+ neurons are required for 2PE attraction. Similarly, the neurons required for TMT aversion could be marked by a gene orthogonal to the observed VGluT gradients, and genetic access to olfactory aversion could potentially be established by investigating these more sparsely expressed marker genes. In both cases, though, further investigation into these other cell types would enhance our understanding of both the plCoA and innate olfactory behavior and allow more precise manipulations in the future.

## Divergent plCoA neurons defined by projection targets control approach and avoidance

In this work, we perform the first comprehensive characterization of plCoA's downstream outputs in the mouse brain. These outputs are dominated by regions generally involved in valence and emotion, such as the NAc, BNST, MeA, BLA, and other amygdalar nuclei, or olfactory areas, such as the pmCoA, OT, and PIR (*Figure 6A–D*). These outputs are consistent with a role for plCoA in motivational valence for odor. The plCoA also appears to form numerous intra-regional connections, where a significant proportion of synapses formed with other neurons are within the region itself. This raises the possibility that plCoA is not simply a feedforward relay but performs local recurrent processing as well. Recurrent networks in other sensory systems expand the dimensionality of encoding schemata and incorporate additional features to generate mixed, continuously updating representations of relevant information (*Singer, 2021*). This raises a number of interesting questions regarding information processing within plCoA. For instance, how does odor representation change based on differences in experience and internal state? Further explorations of information transformation and encoding within plCoA will enrich our understanding of this region.

The NAc and MeA are interesting downstream targets given their known relationships to appetitive and aversive responses, respectively. NAc has historically been critical to the manifestation and processing of reward and motivated behaviors, though this view has been expanded and made more nuanced with a recent focus on action selection (*Floresco, 2015*). On the other hand, MeA has been linked to defensive and stress-related behaviors in response to aversive stimuli. MeA has also been specifically linked to olfactory aversion in past studies, as TMT has previously been shown to activate the MeA (*Day et al., 2004*), which is necessary for TMT-induced defensive behaviors (*Müller and Fendt, 2006*), though the upstream circuits and processing were not yet investigated. These circuits are also notable regarding other features of plCoA spatiomolecular organization, as the projections to the downstream regions of interest are the two that diverge to the greatest extent between aplCoA

and pplCoA, and between plCoA[VGluT2+] and plCoA[VGluT1] glutamatergic neurons. Given the relationship between spatiomolecular patterning and the simple wiring and organizational rules used to structure innate circuits, it would be interesting to investigate the plCoA through the lenses of development and genetic variation. These intersect in recent discussions of genetic bottlenecking, where the genome encodes general rules for circuit organization and development that nevertheless yield specific responses to specific stimuli (*Zador, 2019*). Such networks have numerous theoretical advantages, such as reduced information requirements and higher performance at criterion, providing a conceptual basis for why spatiomolecularly stereotyped circuits yield innate behaviors, and why these innate behaviors are adaptive in naturalistic settings (*Zador, 2019*; *Koulakov et al., 2022*; *Barabási et al., 2023*).

## Neuronal activity in plCoA encodes odor identity via sparse population code

The plCoA receives spatially ordered inputs and has a cell-type-specific topographical organization with divergent outputs that mediate approach and avoidance responses. This anatomical organization is consistent with either labeled-line or divergent paths motifs, although precise connectivity from OB has not yet been described. However, labeled line coding motif is not apparent in neural activity, but instead, there is a sparse population code for odor identity with no apparent valence-specific responses, indicating that the plCoA cannot function as a pure labeled-line relay. Moreover, the absence of valence encoding is distinct from that seen in other divergent path motifs, suggesting the plCoA represents a different circuit model for valence.

We were surprised to find that despite a robust organization of plCoA cell types and projections that support approach and avoidance behaviors, the neurons do not appear to encode appetitive and aversive responses at the single neuron or population level (*Figure 1*). However, when put in the context of the olfactory system, these results are less surprising. Sparse, distributed population encoding appears to be a general feature of all olfactory regions observed thus far, regardless of their specific structure or computational function within olfaction. In the OE, OSNs expressing a single receptor will bind multiple odorants and establish odor identity through a combinatorial code.[21,23] In the OB, axon terminals from OSNs expressing the same receptor converge into specific stereotyped glomeruli, where mitral/tufted cells (M/T) then transmit this information to third-order olfactory regions. Both layers represent the odor identity via a sparse population code in a spatially distributed manner (*Soucy et al., 2009*; *Chae et al., 2019*). The primary olfactory cortex further represents odor identity through sparse, spatially distributed combinatorial population activity in a similar manner (*Stettler and Axel, 2009*). The OT, a striatal region primarily composed of *Drd1*- or *Drd2*-expressing medium spiny neurons, also broadly represents odor identity across both cell types, despite their opposing roles elsewhere in striatum, and a contribution to learned hedonic value is currently debated (*Lee et al., 2023*; *Martiros et al., 2022*). Although direct comparisons of olfactory regions find some

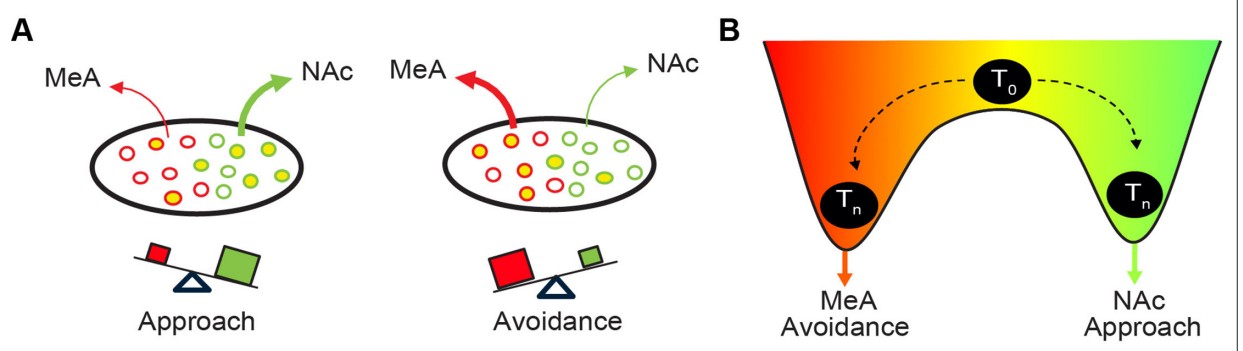

**Figure 8.** Models that could support innate odor-evoked attraction and avoidance. Two potential models that could support valence responses to odor with a population code imposed on divergent circuitry. (**A**) Balance of activation between MeA- and NAc-projecting neurons determines the valence. In this model, an odor may activate a different proportion of these projection-defined neurons, and the valence is determined by the balance. For example, activation of more NAc-projecting neurons should cause attraction, and activation of more MeA-projecting neurons should cause aversion. (**B**) Dynamic activity evolves over time due to recurrent processing or integration of behavioral state variables, in an attractor-like network. In this model, the activity may evolve from an initial broad population code ($T_0$) towards preferential activation of one output population over time ($T_n$).

differences in features like sparsity, clustering, or correlation structure, these are generally reported as differences in degree, not in kind. Given the remarkable similarity in encoding between these structurally and functionally distinct regions, it logically follows that an olfactory region with amygdalar structure would follow similar population encoding principles. Indeed, were plCoA to predominantly encode stimulus valence in a manner akin to other amygdalar regions, it would reflect a greater divergence from olfactory coding principles than what we observe. Instead, the olfactory encoding scheme employed by plCoA reveals a framework for valence encoding that is distinct from that in the extended amygdala.

How does innate valence arise from a population code imposed on stereotyped circuitry? One possibility is that the valence of an odor is defined by the proportion of NAc and MeA-projecting neurons activated, mixing features of divergent paths and opposing components motifs. In this model, a given odor should activate some balance of NAc and MeA projecting neurons, and the proportional balance determines the behavioral response (*Figure 8A*). The activation of TMT-labeled neurons (*Figure 2—figure supplement 2*) in different domains is consistent with this model. That is, if the valence is determined by the balance of activation of NAc and MeA projecting neurons, then activating a subset of that balance in the anterior and posterior should drive the valence-specific response, which is indeed what we observe. If true, one should also be able to predict the valence of the odor by recording from a large sample of these projection-defined neurons. A similar model has also been proposed in the OB (*Qiu et al., 2021*). Another possibility is that activity could evolve over the time course of behavior in freely moving animals such that a mixture of inputs from local circuitry and long-range circuits could function like attractor networks to shape output into one of a few convergent states (*Figure 8B*), consistent with attractor networks in the hypothalamus (*Nair et al., 2023*). A third possibility is that valence may emerge in combination with behavior and internal state. In this model, the plCoA integrates sensory information with other state variables to shift the activity towards one output. If either of these latter two models is correct, recording activity over longer time scales in freely moving animals could provide insight into how activity correlates with behavior. Alternatively, valence signals may arise downstream of plCoA, but molecular mechanisms to support this model are currently undescribed, and such a model would be at odds with our finding of divergent valence-specific circuitry required for attraction and aversion. Moreover, innate behavior is dynamic and more complex than simple attraction and avoidance, and it is possible that the neural activity is as dynamic, changing over the course of behavior. Thus, it will be important to investigate the activity of in the plCoA in behaving animals.

## A unique amygdala circuit motif for valence using a population code

The widespread circuit model for valence encoding centers on divergence, in which information is routed to different pathways depending on the valence of the stimulus. The plCoA circuit we have identified is anatomically similar, but unique in its encoding properties. Most divergent circuits function to generalize stimuli, discarding information about stimulus identity to simplify to low-dimensional valence signals from distinct neuronal populations. In contrast, the plCoA groups stimuli, using a high-dimensional code of odor identity that appears to be routed through divergent projections to mediate opposing responses. In this model, the valence of an odor could be determined based on the population dynamics and composition within a distributed, sparse population code that ultimately funnels information through divergent pathways corresponding to their innate significance. Such a model could serve to increase the flexibility of the system while retaining the ability to yield stereotyped responses.

## Limitations of the study

The odorant, TMT, is a thiol-containing odorant and has long been suspected of acting as an irritant through the trigeminal system and it is known to activate TrpA1 channels in trigeminal neurons (*Wang et al., 2018a*). However, other work has shown that removal of the olfactory bulb or $ZnSO_4$ disruption of OSNs eliminates TMT-evoked freezing, whereas lesion of the trigeminal nerve does not (*Hacquemand et al., 2010*; *Ayers et al., 2013*). Further, in our assay, TMT avoidance requires the olfactory bulb and plCoA circuitry indicating the involvement of the olfactory system in TMT-evoked behaviors (*Root et al., 2014*). Thus, despite the differences in observations in underlying pathways for TMT

responses, the aversion measured in our assay is dependent on olfactory pathways, but there is a potential for it to also engage the trigeminal pathway.

The conclusion that the plCoA circuitry mediates approach and avoidance is limited by the use of only one attractive and one aversive odor in our behavioral experiments. Thus, it remains unclear how these findings generalize to other odors. The fact that exogenous activation of different pathways is sufficient to produce behaviors of opposite valence suggests that there are dedicated plCoA outputs for different behaviors, but the broader necessity remains to be tested. Moreover, it remains unclear how odor activates neurons with distinct projections. Although we did not observe generalized valence responses to odor in calcium imaging experiments, our investigation only concerns the activity of general plCoA ensembles, not of specific neurons. Thus, it remains possible that neurons projecting to the NAc or MeA are more likely to respond to attractive or aversive odors, respectively. However, we did not observe any bias in odor response based on topographical location, and projection-defined neurons are topographically distributed.

The lack of valence encoding may also be an artifact of head fixation, in which mice do not reliably exert the full repertoire of innate behaviors and are unable to use behavior to approach or avoid the odor. Both internal state and environmental context are known to influence sensory processing (*Lin et al., 2019*; *Fanselow and Nicolelis, 1999*; *Fu et al., 2014*), raising the possibility that odor responses are altered under head-fixed conditions, thereby modifying the apparent tuning properties of plCoA neurons. Odor valence itself may also lose behavioral relevance when the animal cannot approach or avoid the stimulus. Future work will benefit from recording neural activity in freely moving animals to determine whether valence-related tuning emerges under more naturalistic conditions.

We provide evidence that plCoA projections to the NAc and MeA are necessary and sufficient for attraction and avoidance. However, an intersectional chemogenetic approach silences the neurons projecting to a given target, not just the specific projection. It is possible that collaterals from MeA-projecting plCoA neurons contribute to the behavior. However, given that NAc and MeA projection neurons do not co-collateralize, it is reasonable to consider the pathways as distinct from each other. Lastly, it remains unclear what the projections from the plCoA to other areas contribute to odor-evoked behavior. For instance, the BNST and LS have been implicated in a variety of functions including stress, arousal, anxiety, and social behaviors. Thus, it is possible that those projections mediate aspects of behavior beyond attraction and avoidance that were not assessed in our assay.

## Materials and methods

**Key resources table**

| Reagent type (species) or resource | Designation | Source or reference | Identifiers | Additional information |
|---|---|---|---|---|
| Strain, strain background (*Mouse*) | C57BL/6 J | Jackson labs | RRID:MGI:3028467 | |
| Strain, strain background (*Mouse*) | VGlut2-cre | Jackson labs | RRID:IMSR_JAX:028863 | |
| Strain, strain background (*Mouse*) | VGlut1-cre | Jackson labs | RRID:IMSR_JAX:023527 | |
| Strain, strain background (*Mouse*) | Arc-CreER[T2] | Jackson labs | RRID:IMSR_JAX:022357 | |
| Transfected construct (*AAV*) | AAV5-hSyn-eYFP | UNC Vector Core | RRID:SCR_023280 | |
| Transfected construct (*AAV*) | AAV5-hSyn-ChR2-mCherry-WPRE-PA | UNC Vector Core | RRID:SCR_023280 | |
| Transfected construct (*AAV*) | AAV5-EF1A-DIO-hChR2(H134R)-eYFP | UNC Vector Core | RRID:SCR_023280 | |
| Transfected construct (*AAV*) | AAV5-EF1A-DIO-eYFP | UNC Vector Core | RRID:SCR_023280 | |
| Transfected construct (*AAV*) | AAV9-hSyn-jGCaMP8s-WPRE | Addgene | RRID:Addgene_162374 | |
| Transfected construct (*AAV*) | AAV2-hSyn-DIO-hM4D(Gi)-mCherry | Addgene | RRID:Addgene_44362 | |

*Continued on next page*

*Continued*

| Reagent type (species) or resource | Designation | Source or reference | Identifiers | Additional information |
|---|---|---|---|---|
| Transfected construct (*AAV*) | AAVretro-hSyn-EGFP-Cre | Addgene | RRID:Addgene_105540 | |
| Transfected construct (AAV) | AAVDJ-hSyn-FLEX-mRuby-T2A-SynEGFP | Provided as a gift From Byungkook Lim | PMID:28689640 | |
| Transfected construct (*AAV*) | AAV5-EF1A-mCherry-IRES-Cre-WPRE | Addgene | RRID:Addgene_55632 | |

## Experimental model and subject details

All procedures at the University of California, San Diego, and Columbia University were performed in accordance with Institutional Animal Care and Use Committee protocols in accordance with NIH guidelines. All mice were provided food and water ad libitum and maintained on a regular 12 hr reverse light/dark cycle at room temperature, with weight, health, and immune status monitored daily and verified to be within normal ranges. Mouse cages were changed regularly based on degree of soiling. All mice were group-housed with randomly assigned littermates prior to surgery and single-housed after surgery. All animals were used in a single experiment each, except for a subset of mice who underwent 4-quad, EPM, and OFT experiments and performed each test in no specific order.

## Subject details for sequencing experiments
### Subject details for single-nucleus sequencing
All mice for snRNA-seq in the study were males on the wild-type C57BL/6 J background (RRID:IMSR_JAX:000664) and received directly from Jackson Laboratories at 6 weeks of age and acclimated to the colony prior to experiments. Animals were single-housed and placed into sensory deprivation 24 hr prior to sacrifice to reduce artifactual immediate early gene expression. Sacrifice was performed at P60±3 days (n=5–10 mice per pool). Sample size was determined based on number of expected nuclei per region per mouse: estimates of expected nuclei were determined empirically, though nuclear recovery was approximately 20% of total based on cellular density estimates from the Blue Brain Cell Atlas. 20,000 nuclei were targeted per combination of assay, condition, and region, which was determined using SCOPIT v1.1.4, allowing potential detection of at least 10 nuclei from 10 rare subpopulations at 0.1% frequency with 95% probability (*Davis et al., 2019*). A total of 50 mice were used for this purpose.

### Subject details for spatial transcriptomics
In the separate study for spatial transcriptomics, APP23 (B6.Cg-Tg(Thy1-APP)3Somm/J; C57BL/6 J background, RRID:IMSR_JAX:030504) non-transgenic (NTG) littermates control mice were housed in light-tight enclosures (*Sturchler-Pierrat et al., 1997*). The mice were given ad libitum food and water access. This study used a total of 17 mice almost equally distributed across sex, of which sections from 11 sagitally bisected the plCoA and were used in downstream analysis. No analysis of sex differences was performed due to inaccessibility of that information on a per-section basis. However, no such differences were apparent from per-section gene expression correlations reported in supplementary information.

## Subject details for calcium imaging, tracing, and activity manipulations
### Subject details for wild-type experiments
All mice for topographic and projection-defined manipulation and tracing experiments, as well as calcium imaging, were males on the wild-type C57BL/6 J background and received directly from Jackson laboratories before 12 weeks of age. After surgery, mice incubated for at least 21 days if injected with virus or at least 7 days if injected with a retrograde tracer (e.g. retrobeads, cholera toxin B) to allow virus to express and tracers to travel in retrograde, respectively. All surgeries and downstream experiments were performed on mice at least 8 weeks of age.

### Subject details for transgenic experiments
We used VGlut2-IRES-Cre (Slc17a6tm2(cre)Lowl; C57BL/6 J background, RRID:IMSR_JAX:028863; *Slc17a6-Cre*) and VGlut1-IRES2-Cre-D (Slc17a7tm1.1(cre)Hze; C57BL/6 J background,

RRID:IMSR_JAX:023527; *Slc17a7-Cre*), Arc-CreER[t2] (RRID:IMSR_JAX:022357) strain mice for molecularly defined optogenetic stimulation experiments and genotype-specific tracing. These mice were bred on-site at UCSD and were genotyped in-house using genomic DNA from ear tissue amplified with the default primer sets listed by Jackson Laboratories. All mice used for experiments had a heterozygous genotype for the transgenic construct of interest. After surgery, mice were incubated for at least 21 days to allow viral expression.

## Methods details

### Stereotactic surgery procedures

All surgeries were performed under aseptic conditions using a model 1900 digital small animal stereotaxic instrument (Kopf Instruments). Mice were initially anesthetized in a sealed box containing 5% gaseous isoflurane, and then deeply anesthetized using isoflurane (2.5% in 1 L/min of $O_2$) during surgeries (VetFlo, Kent Scientific Corporation). We immobilized and leveled the head in a stereotaxic apparatus (Kopf Instruments), removed fur from the scalp by shaving, applied eye lubricant (Optixcare), cleaned the incision site with 70% ethanol and betadine prior to incision, peeled off connective tissue, and dried the surface of the skull prior to craniotomy before proceeding with injections and implantations specific to certain experiments. All virus injections were performed at 2 nL/s using a pulled glass pipette (Drummond) and a Nanoject III pressure injector (Drummond). To prevent backflow, the pipette was left in the brain for 15 min after injection.

### Surgeries for calcium imaging

Surgeries for imaging experiments were performed in a manner similar to that previously described (*Lee et al., 2023*). The skull was prepared with OptiBond XTR primer and adhesive (KaVo Kerr) prior to the craniotomy. After performing a craniotomy 1 mm in diameter centered around the virus injection site, a 27 G blunt needle was used to aspirate ~2.5 mm below the brain surface. 600 nl of AAV9-hSyn-jGCaMP8s-WPRE ($2.5 \times 10^{13}$ gc/ml, Addgene) was diluted to ~$1 \times 10^{12}$ gc/ml gc/ml and injected into the left middle plCoA in two 300 ul boli, one in layer 2 and one 400 µm dorsal (+0.4 DV) in layer 3. Following the viral injection, a head plate (Model 4, Neurotar) was secured to the mouse's skull using light-curing glue (Tetric Evoflow, Ivoclar Group). At least 30 min after viral injection, a 1 mm GRIN lens (NA, ~1.9 pitch, GrinTech) was sterilized with Peridox-RTU then slowly lowered at a rate of 500 µm/min into the craniotomy until it was 200 µm dorsal (+0.2 DV) to the injection coordinate. The lens was adhered to the surface of the skull using Tetric Evoflow. We then placed a hollow threaded post (AE825ES, Thorlabs) to act as a housing for the lens and adhered it using Tetric Evoflow. Any part of the skull that was still visible was covered using dental cement (Lang Dental). Finally, the housing was covered with a nylon cap nut (94922 A325, McMaster-Carr) screwed onto the thread post to protect the lens in between imaging. Animals were left on the heating pad until they fully recovered from anesthesia.

### Surgeries for optogenetic stimulation

For optogenetic topographic- or projection-specific stimulation experiments, we injected wild-type C57BL/6 J mice between 2 and 4 months of age with 100 (if topographic) or 150 (if projection-specific) nL of either AAV5-hSyn-eYFP ($3.3 \times 10^{12}$ gc/ml, UNC Vector Core) or AAV5-hSyn-ChR2-mCherry-WPRE-PA ($4.1 \times 10^{12}$ gc/ml, UNC Vector Core). For Cre-dependent molecularly defined optogenetic stimulation, we injected 200 nl AAV5-EF1A-DIO-hChR2(H134R)-eYFP ($5.5 \times 10^{12}$ gc/ml, UNC Vector Core) or AAV5-EF1A-DIO-eYFP ($4.0 \times 10^{12}$ gc/ml, UNC Vector Core). All such injections were left unilateral, into either aplCoA or pplCoA for topographic photostimulation, and into middle plCoA for projection- or genotype-specific photostimulation. For topographic and genotype-specific photostimulation animals, we implanted the fiber 300 µm (+0.3 DV) directly above the injection site with all other coordinates remaining constant. Anterior-posterior axis positioning arose from stochastic variation in virus and fiber placement. For projection-specific photostimulation animals, we implanted the fiber 300 µm (+0.3 DV) directly above either the MeA or NAc, holding all other coordinates for the two regions constant as described above. For all optogenetic stimulation experiments, we implanted a fiber optic cannula (2.5 mm ferrule outer diameter, 200 µm core, 0.39 numerical aperture; RWD) 300 µm above the targeted stimulation site. These fibers were affixed onto the skull using OptiBond XTR (Kerr) and stably secured with Tetric Evoflow (Ivoclar Vivadent) coated with cyanoacrylate

(Toagosei). After surgery, we injected all mice with 0.04 mL Buprenorphine SR (Ethiqa XR, Fidelis) for pain management. All mice were singly housed immediately after surgery and returned to the colony once ambulatory.

## Surgeries for chemogenetic inhibition

For all plCoA inhibition experiments, we bilaterally injected C57BL/6 J mice between 2 and 4 months of age with 250 nL of either AAV2-hSyn-DIO-hM4D(Gi)-mCherry ($7.1×10^{12}$ gc/ml, Addgene) or AAV2-hSyn-mCherry ($1.8×10^{13}$ gc/ml, Addgene) virus. For projection-specific inhibition experiments, wild-type C57BL6/J mice were used, and AAVretro-hSyn-EGFP-Cre ($1.5×10^{12}$ gc/ml, Addgene) either 50 nL were injected into MeA or 300 µL were injected into NAc. For genotype-specific inhibition experiments, VGluT2-Cre or VGluT1-Cre mice were used. All injections were bilateral and targeted to middle plCoA.

## Surgeries for fluorescent tracing

For non-topographic anterograde tracing experiments, we bilaterally co-injected mixed 50 nl AAVDJ-hSyn-FLEX-mRuby-T2A-SynEGFP ($4.0×10^{12}$ gc/ml, Addgene, a gift from Byungkook Lim) and AAV5-EF1A-mCherry-IRES-Cre-WPRE ($1.9×10^{12}$ gc/ml, UNC Vector Core) into middle plCoA (–1.8 AP, +/-2.9 ML, –5.95 DV). For topographic anterograde tracing experiments, we unilaterally injected 20 nl AAV8-hSyn-hChR2(H134R)-mCherry ($2.1×10^{13}$ gc/ml, Salk GT3 Viral Vector Core) and AAV8-hSyn-hChR2(H134R)-eYFP ($3.2×10^{13}$ gc/ml, Salk GT3 Viral Vector Core) into aplCoA (–1.4 AP, –2.8 ML, –5.95 DV) and pplCoA (–2.1 AP, –3.0 ML, –5.95 DV), counterbalancing region by fluorophore. For retrograde topographic tracing experiments, we unilaterally injected Red Retrobeads IX (Lumafluor) into either MeA (50 nl, –1.2 AP, –2.0 ML, –5.5 DV) or NAc (+1.1 AP, –1.35 ML, –4.5 DV), at volumes of 50 nl or 300 nl, respectively. For anterograde collateralization experiments, AAVretro-EF1A-IRES-Cre ($1.3×10^{13}$ gc/ml, Addgene) was injected into either MeA or NAc, and AAVDJ-EF1A-DIO-hChR2(H134R)-eYFP-WPRE-pA ($4.03×10^{13}$ gc/ml, Salk GT3 Viral Vector Core) was injected into middle plCoA. For genotype-specific anterograde tracing experiments, we injected 50 nl AAVDJ-Ef1a-DIO-ChR2(H134R)-eYFP-WPRE-pA into middle plCoA in VGluT2::Cre or VGluT1::Cre mice.

## Arc-Cre labeling of neurons

For experiments using Arc-CreER[T2] mice, odor-responsive neurons were labeled as previously described (*Root et al., 2014*). One week after the viral injection and fiber implantation, the mice were placed in a clean cage housed in an isolated, quiet, and dark enclosure for 1 day prior to odor exposure. Mice were administered tamoxifen (2 mg/ml in a solution of 10% ethanol and 90% corn oil) by intraperitoneal injection. Six hours after the tamoxifen injection, they were exposed to TMT (1 µL) on a cotton swab placed in their cage for 5 min.

## Calcium imaging

### Odor exposure

Odor exposure for imaging experiments was adapted from methods previously described (*Lee et al., 2023*). Odor was delivered to the mouse using a custom-built olfactometer. Compressed medical air was split into 2 gas-mass flow controllers (Aalborg). One flow controller directed a constant rate of 1.5 L/min to a hollowed-out Teflon cylinder. The other flow regulator was connected to a 3-way solenoid valve (The Lee Co.). Prior to odor delivery, the 3-way valve directed clean air at 0.5 L/min to the Teflon cylinder. During odor delivery, the 3-way valve directs air to an odor manifold, which consists of an array of two-way solenoid valves (The Lee Co.), each connected to a different odor bottle. The kinetics and consistency of odor delivery were characterized using a miniature Photoionization Detector (mPID; Aurora Scientific) mounted above the odor port. The odors were detected using different gain settings on the PID control box as follows: 1 x for TMT and 4MT, 5 x for IAA, 10 x for Peanut, 2PE, and Heptanol. Depending on the trial type, the appropriate two-way valve opens, directing 0.5 L/min of air flow through the odor bottle containing a kimwipe blotted with 20 µl of odorant, except for 2PE and peanut oil, in which 500 µl was used.

Prior to imaging, mice were habituated to the head fixation device (Neurotar) and treadmill for at least 3 days for at least 5 min per session beginning at least 10 weeks after surgery. The treadmill parts were 3D printed using an LCD printer (EPAX) from publicly available designs. Walking behaviors

were measured using a quadrature encoder (Broadcom). A video feed of the animal's face was also recorded using a camera (Basler) with an 8–50 mm zoom lens (Arducam) at 20 Hz with infrared illumination (Lorex Technology). Animals were exposed to the following odors for 5 s: the appetitive odors 2-phenylethanol (Sigma-Aldrich) and peanut oil (Spectrum), the neutral odors 1-heptanol (Sigma-Aldrich) and isoamyl acetate (Sigma-Aldrich), and the aversive odors trimethylthiazoline (BioSRQ) and 4-methylthiazole (Sigma-Aldrich). Within a single contiguous exposure session, each of the six odors was provided 20 times with 12–18 s of inter-trial interval. Trials were organized into 6 blocks, each of which consisted of 20 trials of each of the 6 odors in counterbalanced order without any odors of similar innate valence adjacent to each other in the trial structure.

## Two-photon microscope data acquisition

$Ca^{2+}$ imaging data was acquired using an Olympus FV-MPE-RS Multiphoton microscope with Spectra Physics MaiTai HPDS laser, tuned to 920 nm with 100 fs pulse width at 80 MHz. Each 128×128 pixel scan was acquired with a 20 x air objective (LCPLN20XIR, Olympus), using a Galvo-Galvo scanner at 5 Hz. Stimulus delivery and behavioral measurements were controlled through a custom software written in LabVIEW (National Instruments) and operated through a DAQ (National Instruments). Each imaging session lasted up to 90 min and was synchronized with the stimulus delivery software through a TTL pulse. Animals were excluded from analysis if histology showed that either the GRIN lens or the jGCaMP8s virus was mistargeted or the motion during imaging was too severe for successful motion correction.

## Behavioral assays

Mice had been handled for 5 days prior to experiments and acclimated to the room for an hour prior to testing. All behavioral experiments were performed during the dark period of the light/dark cycle at least an hour away from the switch between the two photoperiods. Not all mice were run in all assays, as elevated plus maze and open field tests were added after a significant proportion of four-quadrant data was collected at targeted sample sizes and mice had already been sacrificed.

For all optogenetic experiments, optical fibers (200 mm, 0.39 numerical aperture, Thorlabs) were epoxied to 2.5 mm stainless steel ferrules (Precision Fibre Products), and polished with a fiber optic polishing kit (Thorlabs) to achieve a minimum of 80% transmission. After surgical implantation, the ferrules protruding from the mouse's head were coupled to an ADR-800A 100 mW 473 nm laser (LaserCentury) via custom-made patch cables with a single rotary joint (Doric Lenses) between the mouse and laser. Laser intensity was set to 5–8 mW at the end of the patch cable. For inhibition experiments, all mice were injected intraperitoneally 60 min prior to the beginning of the behavioral trial with either sterile PBS vehicle or with clozapine-N-oxide (CNO) dihydrochloride (Hello Bio) dissolved in sterile PBS for a dosage of 2 mg/kg.

## Four quadrant open field assay

The four-quadrant open-field task was performed as previously described (*Root et al., 2014*). In short, all behavioral assays took place in a four-quadrant open field chamber. Airflow was pumped into each quadrant via gas-mass flow controllers 150 mL/min (Cole-Parmer). Airflow exited the chamber via a 1-inch outlet in the center of the chamber's floor covered by steel mesh, and the outlet was connected to a vacuum line with a gas-mass controller set to 750 mL/min. The chamber was housed in the dark and illuminated from below by infrared lighting. A Basler A601FM camera (Edmund Optics) mounted above the chamber recorded videos of behavioral trials at 4 Hz, and custom software written in LabVIEW (National Instruments) tracked the position of the mouse in real time for each frame. The symmetrical four-quadrant open field chamber was contained in a lightproof structure (0–10 lux) and illuminated by infrared lights, removing any potential spatial cues available to the animals with respect to the room or its surroundings. In optogenetic experiments, an additional 5 cm spacer was added to the chamber flush with the walls to raise their height for more naturalistic behavior, and an acrylic ceiling with a top with a circular opening 30.5 cm in diameter was added to prevent escape while allowing the fiber optic cable to move freely.

In optogenetic experiments, the laser was pulsed with 50 ms bins at 10 Hz, and there was a steep gradient from 1 to 10 Hz along the perimeter of the quadrant, increasing as proximity to the corner of the quadrant decreased. Preliminary experiments in topographical stimulation animals identified

no clear behavioral effects from the location of the stimulus quadrant itself (data not shown), and all other downstream stimulation experiments exclusively used the lower right quadrant for stimulation to simplify data analysis. The lasers were controlled by TTL modulation from custom LabVIEW software synchronized to the video capture system.

For inhibition experiments, odor was applied by solenoid valves redirecting airflow through 100 mL glass bottles containing 1 μL of a pure odorant on a small piece of Kimwipes. Odorants used were either the previously validated innately aversive 2,5-dihydro-2,4,5-trimethylthiazoline (BioSRQ) or the innately appetitive 2-phenylethanol (Sigma-Aldrich) on a small piece of Kimwipe. All odors were presented in the lower-right quadrant, and all trials were spaced out with at least an hour between runs, during which vacuum was applied to the chamber. Odors and injection treatments were given in counterbalanced, independent order within experimental groups.

## Open field test

The open field is a square arena illuminated to 100–150 lux by ambient lighting. Mice were habituated to the room for at least an hour prior to testing, but otherwise had no prior experience in the arena prior to exposure. Mice were placed in the center of a square arena (27.3×27.3 × 20.3 cm, Med Associates) with four transparent plexiglass walls. Overall locomotion, immobility, and time spent in corners and center regions of arena during each epoch was analyzed for each mouse. Immobility was defined as movement under 0.5 cm/s for a period of at least 1 s, while the center was defined as the middle 13.7×13.7 cm square in the center of the arena and the corners as the corner regions that do not overlap with the center square in either direction (25% of arena area for each region). For optogenetic experiments, mice were allowed to move freely throughout the arena for 25 min total, with 5–8 mW 473 nm light stimulation pulsed with 50ms bins at 20 Hz, alternately delivered during the 5–10 min and 15–20 min epochs (OFF, ON, OFF, ON, OFF). For chemogenetic experiments, mice were allowed to freely move through the area for 10 min total.

## Elevated plus maze

The arms of the elevated plus maze were 30.5×5.5 cm. The height of the closed arm walls was 15 cm. The maze was elevated 40 cm from the floor and was placed in the center of the behavior room away from other stimuli. Arms were illuminated to 0–10 lux, with infrared illumination. Mice were placed in the center of the maze at the beginning of each trial. For optogenetic experiments, mice were allowed to move freely throughout the maze for 15 min total, with 5–8 mW 473 nm light stimulation pulsed with 50ms bins at 20 Hz delivered during the 5–10 min epoch (OFF, ON, OFF). For chemogenetic experiments, mice were allowed to freely move through the area for 10 min total.

## Histology
### Section preparation

All sacrifices were performed during the dark period of the light cycle. Animals were anesthetized prior to sacrifice via combined intraperitoneal injection of 150 mg/kg ketamine (Zetamine, Vet One) and 15 mg/kg xylazine (AnaSed, AMRI Rensselaer). Except for animals used in sequencing studies, animals were subject to transcardial perfusion with 10 mL of sterile phosphate-buffered saline (PBS), followed by 10 mL 4% paraformaldehyde (PFA) solution. The brain was then extracted from the animal and placed into a 4% (PFA) solution in PBS for at least 36 hr until it was sectioned on a VT1000S vibratome (Leica). For tissue extracted for non-RNAscope histology, mice were transcardially perfused with 20 ml phosphate buffered saline (PBS) followed by 20 ml 4% paraformaldehyde (PFA) in PBS. All brains were extracted and post-fixed for at least 24 hr in 4% PFA. For tissue extracted for RNAscope, mice under 6 months of age were decapitated once unconscious and their brains were extracted into a square Peel-A-Way embedding mold (Polysciences) and embedded in OCT (Fisher), and then snap-frozen in a dry-ice/isopentane slurry and stored at –80 °C until cryosectioning within a month of sacrifice.

Tissue was mounted in 5% agarose and sectioned sagittally on a vibratome for retrograde experiments, or sectioned coronally without mounting for all other non-RNAscope experiments. These sections were cut at 50 μm and stored in PBS before processing. All connectomic quantitation was performed on samples using epifluorescence without immunolabeling to avoid potential bias due to non-stoichiometric antibody binding, while all others were immunolabeled for visualization of viral

targeting accuracy and collection of representative images. Immunolabeling of eYFP and other GFP-derived fluorophores was performed using goat anti-GFP primary antibodies (Abcam) and Alexa Fluor 488-conjugated anti-goat secondary antibodies (Invitrogen), while immunolabeling of mCherry and other DsRed-derived fluorophores used rabbit anti-DsRed primary antibodies (Takara) and Alexa Fluor 588-conjugated anti-rabbit antibodies (Invitrogen), all diluted 1:1000 in PBS-T. All non-RNAscope sections were mounted on Superfrost Plus microscope slides (Fisher) and counterstained with Fluoro-mount-G containing DAPI (SouthernBiotech). Sections were stored long-term at 4 °C.

### RNAscope fluorescence in situ hybridization

RNAscope sections were cut at 15 μm on a CM 1950 cryostat (Leica) and mounted on Superfrost Plus slides and stored at –80 °C until processing via RNAscope within a month of mounting. RNAscope was performed in an RNA-free environment according to manufacturer instructions using the Multiplex Fluorescent Reagent Kit v2 (Advanced Cell Diagnostics). RNAscope was performed using the probes *mm-Slc17a7* (VGluT-1) in the C2 channel, and *mm-Slc17a6* (VGluT2) in the C3 channel, dyed with Opal 520 and Opal 690 in a counterbalanced manner at 1:15,000 concentration to reduce background fluorescence and allow quantitation of unsaturated, clearly-distinguishable puncta. Processed RNAscope sections were then mounted with Prolong Antifade Diamond (Thermo Fisher) and stored long-term at 4 °C.

### Fluorescence image acquisition

Non-RNAscope images were acquired at ×10 magnification with an VS120 slide scanner (Olympus), with settings held constant within experiments. Confocal fluorescence images for RNAscope were acquired on an SP8 (Leica) confocal laser scanning microscope using a 40 x/1.30NA oil immersion objective. Serial Z-stack images were acquired using the LASX software at a thickness of 1 μm per Z stack, with 14–21 planes taken per image. Images were acquired with identical settings for laser power, detector gain, and amplifier offset for each set of counterbalanced probe-fluorophore combinations.

## Sequencing data acquisition

### Tissue extraction and cryopreservation for single-cell sequencing

Once unconscious, mice animals were transcardially perfused with ice-cold, carbogen-bubbled (95% O2, 5% CO2), nuclease-free, 0.22 μm sterile-filtered artificial cerebrospinal fluid (ACSF) with a composition of 93 mM N-methyl-D-glucamine, 2.5 mM KCl, 1.2 mM NaH2PO4, 30 mM NaHCO3, 20 mM HEPES, 25 mM glucose, 5 mM sodium ascorbate, 2 mM thiourea, 3 mM sodium pyruvate, 13.2 mM trehalose, 12 mM N-acetyl-cysteine, 0.5 mM CaCl2, 10 mM MgSO4, and 93 mM HCl, at pH 7.3–7.4 (*Tasic et al., 2018*). Following transcardial perfusion, brains were immediately extracted and submerged into ice-cold carbogen-bubbled ACSF, with less than 5 min between the beginning of perfusion and final submersion after extraction. Brains were serially sectioned in ice-cold, carbogen-bubbled ACSF on a VT1000S vibratome (Leica) with polytetrafluoroethane-coated razor blades (Ted Pella) at 0.15 mm/s and 100 Hz, dividing the whole cerebrum into 400 μm coronal slices. Target regions were microdissected from these slices under a stereomicroscope using a sterile blunt-end needle (22 gauge for CeA, ASt, and tail of striatum, 16 gauge for dorsal striatum). All regions were targeted using Paxinos & Franklin, 5th Edition, as reference (*Paxinos and Franklin, 2019*). Extracted tissue samples were recovered in ice-cold, nuclease-free, 0.22 μm sterile-filtered cryoprotective nuclear storage buffer, composed of 0.32 M sucrose, 5 mM CaCl2, 3 mM magnesium acetate, 10 mM Trizma hydrochloride buffer (pH 8.0), 1 mM dithiothreitol, 0.02 U/μl SUPERase•In RNAse Inhibitor (Invitrogen), and 1 X cOmplete Protease Inhibitor Cocktail with EDTA (Roche). Tissue was then snap frozen using a metal CoolRack M90 (Biocision) pre-chilled to –80 °C and stored at –80 °C until nuclear isolation. Following extraction of tissue regions of interest, remaining portions of sections were fixed in 4% paraformaldehyde and 4',6-diamidino-2-phenylindole (DAPI) was applied to sections at 1 μg/ml. After fixation and staining, sections were mounted and imaged on a VS120 slide scanner (Olympus). From these images, dissection accuracy was assessed for each region, and individual samples were only selected for downstream nuclear isolation if the extracted tissue fell entirely within the defined target regions.

Nuclear isolation procedures were adapted from multiple methods described previously (*Preissl et al., 2018*; *Krishnaswami et al., 2016*). All procedures were performed on ice, and all solutions were

ice-cold, nuclease-free, and 0.22 μm sterile-filtered. Cryopreserved tissue pieces were slow-thawed by incubation at 4 °C for 1 hr prior to isolation. Tissue pieces were then pooled and resuspended in nuclear isolation medium composed of 0.25 M sucrose, 25 mM KCl, 5 mM MgCl2, 10 mM Trizma hydrochloride buffer (pH 7.4), 1 mM dithiothreitol, 0.04 U/μl RNasin Plus RNAse Inhibitor (Promega), 1 X cOmplete Protease Inhibitor Cocktail with EDTA (Roche), and 0.1% Triton-X. The pooled tissue pieces in nuclear isolation medium were transferred to a 2 mL Dounce tissue grinder. Tissue was homogenized by 5 strokes from the loose pestle and 15 followed by the tight pestle, and the resulting homogenate was filtered through a 40 μm Flowmi cell strainer (Bel-Art) into a 1.5 ml Lo-Bind tube (Eppendorf). The homogenate was then centrifuged with a swinging bucket rotor at 4 °C and 1000 x $g$ for 8 minutes. Nuclei were then washed with nuclear flow buffer composed of DPBS with 1% bovine serum albumin, 1 mM dithiothreitol, and 0.04 U/μl RNasin Plus RNAse Inhibitor (Promega) and centrifuged at 4 °C and 500 x $g$ for 5 min, which was subsequently repeated. Nuclei were finally resuspended in nuclear flow buffer containing 3 μm DRAQ7 (Cell Signaling Technology) and again filtered through a 40 μm Flowmi cell strainer into a 5 ml round-bottom polystyrene tube. Each isolation took under 45 min to perform, from homogenization to final suspension.

## Fluorescence-activated nuclei sorting (FANS)

FANS was carried out on a FACSAria II SORP (BD Biosciences) using a 70 μm nozzle at 52 PSI sheath pressure. For FANS, debris was first excluded by gating on forward and side scatter pulse area parameters (FSC-A and SSC-A), followed by exclusion of aggregates (FSC-W and SSC-W), and finally gating for nuclei based on DRAQ7 fluorescence (APC-Cy7). Nuclei were successively sorted into 1.5 ml LoBind tubes (Eppendorf) under the purity sort mode. The tube contained 10 X RT master mix without RT Buffer C. 16,000 total nuclei were targeted for downstream processing, and to account for cytometer errors and subsequent loss of nuclei, 21,000 were sorted into the tube. Nuclei were then immediately processed for snRNA-seq. FANS conditions were optimized for isolation of debris-free nuclei using the LIVE/DEAD Viability/Cytotoxicity Kit for Mammalian Cells (Molecular Probes), adding to the final suspension according to manufacturer instructions and examining on a hemocytometer using an EVOS FL Cell Imaging System (Thermo Fisher) for enrichment of ethidium homodimer-1-positive nuclei and the absence of Calcein AM-labeled cellular debris.

## Tissue extraction and sample preparation for spatial transcriptomics

Mice were euthanized with $CO_2$ followed by decapitation, either in the dark or in the light. Brain hemispheres were collected and placed in OCT and then flash frozen in isopentane in liquid nitrogen. One hemibrain from each mouse was cryosectioned at –18 °C sagittally to a thickness of 10 mm (~2.8 mm from the midline) using a standard Leica CM1860 cryostat and processed according to the recommended protocols (Tissue optimization: CG000240 Visium 10 X Genomics; Gene expression: CG000239). The tissue was immediately mounted on a Visium spatially barcoded slide (10 X Genomics). The tissue was covered with OCT and kept at –80 °C until it was cryosectioned again starting at the same position to a thickness of 10 mm and mounted onto a Superfrost plus microscope slide (Fisherbrand) for staining. Each section covered approximately 80% of the 5000 total spots within their fiducial frame. Slides were stored at –80 °C until use.

## Library preparation

### Library preparation for single nucleus sequencing

Nuclear suspensions were converted into barcoded snRNA-seq libraries using the Chromium Next GEM Single Cell 3' v3.1 Reagent Kits v3.1 Single Index (10 X Genomics). Library preparation for both assays was performed in accordance with the manufacturer's instructions. 10,000 nuclei were targeted during each snRNA-seq library preparation run. 10 X libraries were first sequenced at low depth on a NextSeq 550 Sequencing System (Illumina) to estimate quality and number of nuclei for each library, followed by deep sequencing on a NovaSeq 6000 Sequencing System. All runs were performed using 2x100 bp paired-end reads, outputting data in 28/8/91 bp read format for snRNA-seq runs.

### Library preparation for Visium spatial transcriptomics

Tissue was harvested from 7-month-old, non-transgenic littermates from a cross to an App23 Alzheimer's model in a C57BL6/J background (*Gelber et al., 2026*). Visium spatial gene expression slides and

reagents were used according to the manufacturer instructions (10 X Genomics). Each capture area was 6.5 mm x 6.5 mm and contained 5000 barcoded spots that were 55 μm in diameter (100 μm center to center between spots) providing an average resolution of about 1–10 cells per spot. Optimal permeabilization time was measured at 24 min. Libraries were prepared according to the Visium protocol (10 X Genomics) and sequenced on a NovaSeq4 (Illumina) at a sequencing depth of 182 million read pairs. Sequencing was performed with the recommended protocol in a 28/10/10/100 bp read format. H&E (Hematoxylin, Thermo; Dako bluing buffer, Dako; Eosin Y, Sigma) staining and image preparation was performed according to the Visium protocol. H&E-stained sections were imaged using a Nanozoomer slide scanner (Hamamatsu). Spatial gene expression assay was performed according to the protocol CG000239.

## Quantification and statistical analysis

### Statistical analysis

All statistical details can be found in the figure legends. Sample sizes for behavioral studies were chosen based on past optogenetic studies for each behavior, which had used 6–15 animals per group. Blinding experimenters was not possible for behavioral, imaging, or sequencing experiments, given familiarity with subjects, but manual quantitation for connectivity experiments was performed blinded to group with random assignment. All statistical tests were performed in R (v4.2.3) unless otherwise specified. All statistical tests were performed with two tails. Group comparisons were made using two-way analysis of variance (ANOVA) followed by Bonferroni post hoc tests, except where otherwise specified. All behavioral experiments were performed by multiple experimenters across multiple cohorts each composed of multiple litters, with littermates distributed across control and treatment groups, with all such cohorts yielding similar results (data not shown), and topography stimulation experiments were performed across multiple facilities and institutions. Numbers of mice used for all non-sequencing experiments are reported within the relevant figures, figure legends, and the text.

### Calcium imaging data analysis

#### Image processing

Data analysis for calcium imaging experiments was adapted from methods previously described (*Lee et al., 2023*). Ca$^{2+}$ imaging data were first motion-corrected using the non-rigid motion correction algorithm NoRMCorre (*Pnevmatikakis and Giovannucci, 2017*). Afterward, neural traces were extracted from the motion-corrected data using constrained nonnegative matrix factorization (CNMF) (*Pnevmatikakis et al., 2016*). Spatial components identified by CNMF were inspected by eye to ensure they were not artifacts. A Gaussian Mixture Model (GMM) was used to estimate the baseline fluorescence of each neuron. To account for potential low-frequency drift in the baseline, the GMM was applied along a moving window of 2500 frames (500 s). The fluorescence of each neuron at each time point t was then normalized to the moving baseline to calculate $\Delta F/F = F_t - F_{baseline} / F_{baseline}$. All subsequent analysis was performed using custom code written in MATLAB (R2022b).

#### Hierarchical clustering of pooled averaged responses

ΔF/F in response to all six odors was averaged across trials, then Z-scored. The resulting trial-average values from the following time bins were averaged across time: the first second during each odor, the last second during each odor, and the first second after each odor. The resulting 18-element vectors were sorted into 6 clusters after agglomerative hierarchical clustering using Euclidean distance and Ward linkage.

#### Responsiveness criteria

To determine how many neurons were responsive to a given odor, we compared ΔF/F at each frame during the 2 s odor period against a pooled distribution of ΔF/F values from the 2 s prior to odor onset using a Wilcoxon rank sum test. The resulting p-values were evaluated with Holm-Bonferroni correction to ensure that familywise error rate (FWER) was below 0.05. We also counted the number of neurons that were significantly responsive for at least 4 frames during the odor period to report the total percentage of responsive neurons during odor.

## Single-neuron six-odor classifiers

To test how reliably a single neuron's fluorescence could discriminate between the six odors, we assessed the performance of multinomial regression (MNR) classifiers trained on a single neuron's responses to six odors. For each neuron and odor pair, we averaged the ΔF/F during the last second of the odor exposure for each trial, then Z-scored across all trials. The last second was used because classifier performance is highest during the last second of odor exposure and the first second after odor offset. The resulting feature vector, a single-dimensional vector of length 120 (6 odors x 20 trials), was used to train the MNR classifiers. The fivefold cross-validated accuracies are reported and plotted as violin plots. As a control for each neuron, 100 shuffled classifiers were trained on the data with the odor labels randomly assigned.

## Pairwise Euclidean distance

To quantify the differences among population-level responses to the 6 odors, we quantified the pairwise Euclidean distance between the trajectories of odor responses. First, we subtracted the ΔF/F values during the 2 s prior to odor delivery from each frame, then averaged these values across trials for each odor. The pairwise Euclidean distance at each frame was computed for each odor pair and normalized to the maximum pairwise distance measured in all odor pairs at any time bin. These calculations were carried out separately for each animal and then averaged across biological replicates to report the mean and the standard error.

## Population classifiers

To assess the discriminability of odor responses in high-dimensional space, we measured the accuracy of error-correcting output codes (ecoc) classifiers. At each time point relative to odor delivery, we pooled ΔF/F values from all trials during which either odor was presented. The feature vector is a multidimensional vector of length 120 (6 odors x 20 trials) and varying width k between 1 and 300. These values were then normalized and used to train a multinomial ecoc classifier using a Support Vector Machine (SVM). The accuracy of the classifier was evaluated via 5-fold cross-validation. To compare the classifier accuracies across different numbers of neurons used for training, we randomly selected varying numbers of pooled neurons and used the ΔF/F values during the last second of odor exposure for training.

## Behavioral data analysis

Behavioral metrics (i.e. performance index, port distance, center distance, open field time, and total distance) for the four-quadrant preference test, open field test, and elevated plus maze were calculated on sets of coordinates created by identifying the centroid of the mouse in real time in video collected from an overhead camera (Basler) at 4 Hz using custom Labview code and outputting the centroid's coordinates for each frame. The mouse was automatically identified by taking a background grayscale image of the behavioral assay's environment at the beginning of each trial and detecting shapes of a minimum size deviating from the background image by a specific threshold. The centroid was then determined by automated generation of a bounding box for the mouse in each frame in real time and recording the coordinate of the centroid of this rectangle.

## Four-quadrant task data analysis

Mice were tested as previously described (*Root et al., 2014*). Mice were placed in the chamber for 25 min experiments and tested no more than once per day. The first 10 min served as a baseline test for spatial or temporal bias within the chamber during the trial, and no stimulus of any sort was provided, while the last 15 min were the test of the manipulation. 15 min was chosen to balance time courses of odor responses observed in previous experiments, where appetitive odors tend to yield initial responses that decay, while aversive odors tend to yield responses that grow in magnitude over time. To ensure effects did not arise from ceiling or floor effects in the baseline or from a nonstandard baseline internal state, the mouse had to remain within the stimulus quadrant during the baseline test between ~20–30% of the time or else the experiment was terminated, and the mouse was tested again later. The first 2 min of data after the stimulus was introduced were excluded from the analysis to reduce variance and account for novelty of the stimulus without affecting the overall valence of the behavioral response, and the last minute of data was excluded to ensure no minor differences in

frames captured could affect analysis. For chemogenetic odor response silencing experiments, animals with responses below an absolute value of 0.1 during saline trials were excluded from experiments to avoid false negatives from attempting to silence a response that was not observed at baseline, which applied to less than a quarter of overall animals tested across experimental conditions. Performance index represents the percent difference from chance occupancy in the manipulation quadrant, calculated as PI = $(P - 0.25) / 0.25$; where $P$ is the fraction of time the animal spends in the manipulation quadrant. Mean port distance represents the mean distance of each point to the deepest point into the manipulation quadrant observed at baseline.

## Open field test data analysis

For elevated plus maze analysis, all chemogenetic inhibition trials used the entirety of the 10 min test length as a single period, while optogenetic stimulation trials used the mean of the three 'OFF' periods to compare to the mean of the two 'ON' periods. Three metrics of interest were calculated. Center time was calculated as the proportion of time spent in the middle square of the open field comprising 50% of its total area. Corner time was calculated as the proportion of time spent in the corner squares bounded by the walls and the lines bounding the center region. Time immobile was calculated as the proportion of time when the animal moved less than 1 cm/s for at least a one-second period. The location of the open field and bounding regions was kept constant from trial to trial by registering the apparatus to a bounding box with the same top-down dimensions, and every measured centroid outside of the registered region resulting from shadows cast and other artifacts was interpolated between the closest points before and after within the region.

## Elevated plus maze data analysis

For elevated plus maze analysis, all chemogenetic inhibition trials used the entirety of the 10 min test length as a single period, while optogenetic stimulation trials used the mean of the two 'OFF' periods to compare to the 'ON' period. Three metrics of interest were calculated. Time in the open arms was calculated as the proportion of time spent in the open arms compared to the whole period of interest and did not include time in the center between the two arms. Open arm entries measure the number of episodes where the centroid is observed outside of the bounds of the closed arms or the center region, without any minimum time or distance out onto the open arms. Finally, distance was simply calculated as the distance traveled during each period of interest. Location of open and closed arms was kept constant from trial to trial by registering the apparatus to a cross-shaped bounding box with the same top-down dimensions, and every measured centroid outside of the registered region due to factors like the mouse leaning over the edge of the open arms, among others, was interpolated between the closest points before and after within the region.

## Analysis of single-nucleus RNA sequencing data

### Sequence alignment

All samples were processed using Cell Ranger (v5.0.0; *Zheng et al., 2017*). All processing was done by using Cell Ranger's implementation of STAR to align sample sequence reads to their pre-built mm10 vm23/Ens98 reference transcriptome index 2020 A, with predicted and non-validated transcripts removed. All sequencing reads were aligned to both the exons and the introns present in the index. Samples were demultiplexed to produce a pair of FASTQ files for each sample. FASTQ files containing raw read sequence information were aligned to the Cell Ranger index using the cellranger count command with `--chemistry SC3Pv3` and `--include-introns` flags enabled. Cell Ranger corrected sequencing errors in cell barcodes to pre-defined sequences in the 10X v3 single-index whitelist within Hamming distance 1. PCR duplicates were removed by selecting unique combinations of corrected cell barcodes, unique molecular identifiers, gene names, and location within the transcript. Raw unfiltered count data was read into R (v4.2.1) using the Seurat package (v4.2.0) (*Butler et al., 2018*). The final result of the pipeline was a barcode x gene expression matrix for further analysis downstream.

### Quality control

We used the raw, unfiltered matrix output from Cell Ranger as the input to the beginning of the pipeline. However, to apply a more stringent filter, the emptyDrops dirichlet-multinomial model from the

DropletUtils package (v1.10.2) was applied to each library individually. Droplets with less than 100 total counts were used to construct the ambient RNA profile and an FDR threshold below 0.001 was used to select putatively occupied droplets. All barcodes with greater than 1000 UMIs were further assumed non-empty. Most quality filtration choices were heavily influenced by the recommendations presented in pipeComp (*Germain et al., 2020*). All quality control was performed on each library individually prior to merging. Minimal quality filtering for each barcode was performed by setting a floor of 1000 features per barcode for downstream inclusion to ensure the dataset is entirely composed of high-quality nuclei. Next, to remove highly likely multiplet barcodes, barcodes were filtered out if their count depth was more than 5 median absolute deviations above the median count depth. Barcodes were then removed if their proportion of ribosomal or mitochondrial reads was more than 5 interquartile ranges above the 75th percentile (median absolute deviations cannot be used, because in many cases the median absolute deviation is 0). Heterotypic doublets were identified by creating simulated artificial doublets in scDblFinder (v1.4.1), which uses a DoubletFinder-like model to remove barcodes similar to simulated doublets, with an assumed doublet rate of 1% per 1000 nuclei in the library (*McGinnis et al., 2019*). Scatter (v1.18.3) was used to produce initial diagnostic tSNE and UMAP plots for visually checking the influence of each above metric on the structure of the data (*McCarthy et al., 2017*).

## Data processing/transformation

All datasets (initially for all nuclei and again for selected subclusters) were formatted into Seurat objects (v4.0.0), merged, and then normalized and transformed individually using the SCTransform (v2) variance stabilizing transform, which performs best according to prior comparisons in pipeComp (*Germain et al., 2020*). Following the merge, all genes expressed in three or fewer nuclei of interest were removed from analysis. SCTransform was run returning Pearson residuals regressing out mitochondrial gene expression, retaining the top 5000 highly variable features. The dimensionality of the dataset was first reduced using principal component analysis, as implemented in Seurat's RunPCA function, retaining the top 50 principal components. Principal components were selected for downstream use by using the lower value of either the number of principal components where the lowest contributes 5% of standard deviation and all cumulatively contribute 90% of the standard deviation, or the number of principal components where the percent change in variation between the consecutive components is lower than 0.1%. These principal components were used as input to the non-linear tSNE and UMAP dimensionality reduction methods as implemented by Seurat's RunTSNE and/or RunUMAP functions with 1000 epochs at 0.5 minimum distance, with otherwise default settings.

Clusters were identified via Leiden clustering in latent space using the previously selected principal components as input. Optimal clustering resolution was identified in a supervised manner using clustree, finding the highest resolution for each dataset where clustering remains stable (*Zappia and Oshlack, 2018*). Cluster annotation was performed in a semi-hierarchical semi-supervised manner, where known marker genes were first used to separate all nuclei into neuronal and non-neuronal cell types, and then these cells were re-analyzed and neurons were respectively separated into glutamatergic and GABAergic neurons, while non-neuronal cells were separated into astrocytes, microglia, macrophages, oligodendrocytes and their precursors/lineage, mural cells, endothelia, and vascular leptomeningeal cells. This lower level of cells was then reanalyzed, and novel cell types were then identified within these more-granular known cell types. For identification of known cell types, clusters expressing the same marker genes were manually merged to ensure all cells of a known type were analyzed together, which did not occur for novel cell type identification. Clusters resulting from specific differences in nuclei quality instead of true changes in gene expression (i.e. markedly lower mean UMI/features per nucleus, increased ribosomal/mitochondrial gene expression proportion) were removed prior to final clustering. Relationships between cell type proportion and plCoA zone were quantitated using propeller, treating each library as an independent replicate (*Phipson et al., 2022*).

## Differential expression

Marker genes were identified using the Wilcoxon rank-sum test as implemented by the FindConservedMarkers function in Seurat, using the region as a grouping variable. Genes were accepted as differentially expressed with a minimum proportion cutoff at 0.1 and minimum fold change at 1.5-fold (log2-fold change of 0.585), with a p-value cutoff of 0.01 after Bonferroni correction. To identify

genes differentially expressed by region, single-cell values were converted to pseudo-bulk by batch using the run_de function as implemented in the Libra package (v1.0.0) using default settings with a minimum proportion cutoff at 0.1, and tested for differential expression using edgeR's likelihood ratio test. Zone-specific gene expression was identified by comparing batches from the two isolated zones.

## Analysis of spatial transcriptomics data
### Sequence and image alignment

All samples were processed using Space Ranger (v1.3.0). All processing was done by using Space Ranger's implementation of STAR to align sample sequence reads to their pre-built mm10 vm23/Ens98 reference transcriptome index 2020 A, with predicted and non-validated transcripts removed, as in snRNA-seq data alignment. Samples were demultiplexed to produce a pair of FASTQ files for each sample. FASTQ files containing raw read sequence information were aligned to the index using the spaceranger count command. Space Ranger corrected sequencing errors in cell barcodes to pre-defined sequences in the single-index whitelist within Hamming distance 1. PCR duplicates were removed by selecting unique combinations of corrected cell barcodes, unique molecular identifiers, gene names, and location within the transcript. Imaging data was processed using automatic fiducial alignment and tissue detection on a brightfield input.

### Data processing/transformation

We used the image-filtered matrix output from Space Ranger as the input to the beginning of the pipeline. In a similar manner to snRNA-seq data, all datasets were formatted into Seurat objects (v5.0.0), merged, and then normalized and transformed using the SCTransform (v2) variance stabilizing transform. SCTransform was run returning Pearson residuals regressing out mitochondrial gene expression, retaining the top 5000 highly variable features. The dimensionality of the dataset was first reduced using principal component analysis, as implemented in Seurat's RunPCA function, retaining the top 50 principal components, all of which were used in downstream processing. These principal components were used as input to the non-linear tSNE and UMAP dimensionality reduction methods as implemented by Seurat's RunTSNE and/or RunUMAP functions with 1000 epochs at 0.2 minimum distance, with otherwise default settings. Clusters were identified via Leiden clustering in latent space using all 50 principal components as input. Optimal clustering resolution was identified in a supervised manner using clustree, finding the highest resolution for each dataset where clustering remains stable, choosing a resolution of 0.7. Cluster annotation was performed in a semi-supervised manner, observing where in captured plCoA regions each cluster's spots localized to. For clusters that could not be annotated from spatial location alone (e.g. OLG), marker genes were examined to determine the molecular identity of relevant spots. Spatial data was projected onto neuronal molecular cell types from snRNA-seq data and cell type likelihood was predicted using Seurat's FindTransferAnchors and TransferData functions using snRNA-seq data as a reference and plCoA spatial data as the query, using all 50 PCs. Prediction of a minority of subtypes failed, likely due to low abundance in tissue and/or due to mediolateral spatial differences, alluded to in a separate study, causing the section not to intersect with the part of the tissue containing the relevant neuronal subtypes. Glutamatergic and GABAergic molecular subtype likelihoods were predicted separately to remove noise and increase modeled prediction confidence.

## Histological image analysis
### Registration and localization

Histology for all animals and samples was examined prior to inclusion. Localization within the coronal plane was determined by registering the coronal slice to the Allen Brain Atlas via the ABBA plugin (*Chiaruttini et al., 2025*), using elastix to sequentially perform affine and spline registration of the DAPI channel of the slice to the Nissl channel of the atlas. The region of interest was then compared to the Paxinos and Franklin atlas to confirm localization and find the region's anteroposterior distance from bregma. This combined method was used because sections cannot be accurately registered to the Paxinos and Franklin atlas due to low Z-resolution, while the Allen Brain Atlas lacks information about anteroposterior distance from bregma. Exclusion based on histology would occur when most of the intervention fell outside of the region of interest. Due to these differences, individual

representative images use the individually registered Allen Reference Atlas schematics with the comparable Paxinos and Franklin anteroposterior coordinates noted, while consolidated targeting schematics use the Paxinos and Franklin atlas for visualization.

## Quantification of histological fluorescence

In anterograde tracing experiments, output quantification was performed based on background-corrected total fluorescence. For all non-collateralization anterograde experiments, fluorescence intensities were quantified using FIJI (v2.9.0) throughout the whole brain in a series of evenly spaced 50 μm coronal sections, manually segmenting by region with all settings held constant within experiments. For collateralization experiments, we exclusively examined fluorescence in the MeA and NAc. We calculated background-corrected total fluorescence using the equation $F_{total} = ID - (Area \times F_{background})$, where $F_{total}$ is the background-corrected total fluorescence, $ID$ is the integrated density, and $F_{background}$ is the mean background fluorescence measured from four randomly selected areas per section not receiving input from plCoA. Overall proportion was calculated by taking the sum of background-corrected fluorescence values across all sections for a given region and dividing it by the sum of all background-corrected values. For retrograde experiments, we quantified the number of cells using the Cell Counter plugin (v3.0.0) in FIJI. The sagittal brain slices containing the plCoA were then compared to Paxinos and Franklin, 5th Edition to count the number of cells found per distance away from bregma from −1.3 to −2.5 mm in increments of 100 μm. At least two sections per region per animal were analyzed. Representative images were produced from slide scanner image output, with background subtraction and uniform brightness and contrast thresholds applied equally to all fluorescent channels in FIJI to avoid potential distortion of visible fluorescence levels.

## Quantification and analysis of RNAscope images

RNAscope images were analyzed as previously described. Images were opened in FIJI and individual Z-planes encompassing the entire ROI were selected from each image for further image processing. Background was subtracted from all channels in all images using the subtract background feature. Masks of each region were drawn based on the mouse brain atlas, and images were then saved as 8-bit TIFFs for further cell and puncta identification in CellProfiler (v4.2.4; *Carpenter et al., 2006*). Image TIFFs were run through CellProfiler using an optimized version of the CellProfiler Colocalization pipeline. The pipeline was optimized to identify DAPI-labeled cells (15–45 pixels in diameter) and then subsequently identify mRNA puncta (4–10 pixels in diameter). DAPI cell detection was further restricted by shrinking DAPI ROIs by 1 pixel. Puncta overlapping with DAPI-identified cells (using the relate objects module) were considered for analysis to assess the level of mRNA expression per cell. To determine if cells were expressing mRNA, a threshold of five or more puncta within twice the diameter of nucleus centered over the nucleus was used. Total number and density of VGluT2+ and VGluT1+ cells in each region of interest were calculated from CellProfiler.csv outputs using custom R scripts.

## Resource availability

### Lead contact

Further information and requests for resources and reagents should be directed to and will be fulfilled by the Lead Contact, Cory M. Root (cmroot@ucsd.edu).

### Materials availability

This study did not generate new unique reagents.

## Acknowledgements

We thank the entire Root lab for helpful discussion and support. We thank B Lim for provision of the AAVDJ-hSyn-Flex-mRuby-T2A-SynEGFP virus. We thank G Pekkurnaz, N Spitzer, and M Pratelli for reagents and facilities support. We also thank C O'Connor and L Boggeman at the Salk flow cytometry core, and N Hah at the Salk next-generation sequencing core. We thank D Jimenez for assistance with histology and colony management. This publication includes data generated at the UC San Diego IGM Genomics Center utilizing an Illumina NovaSeq 6000 that was purchased with funding from a National

Institutes of Health SIG grant (#S10 OD026929). This work was generously supported by the National Defense Science and Engineering Fellowship (JRH), CIHR Postdoctoral Fellowship (FM), JPB Foundation, the PIIF, PNDRF, JFDP, New York Stem Cell Foundation, Klingenstein Foundation, McKnight Foundation, Salk Institute, Howard Hughes Medical Institute, Clayton Foundation, Kavli Foundation, Dolby Family Fund (KMT), the Hellman Fellowship (CMR), and the National Institutions of Health, via grants through the NIA (RF1AG061831-01S1, PAD), NIMH (K99MH121563-02, FM; R01MH115920-03 and R37MH102441, KMT), NIDDK (DP2DK102256-01S1, KMT), and NCCIH (DP1AT009925-04, KMT), and the NIDCD (R00DC014516-05 and R01DC018313-01A1, CMR).

## Additional information

### Funding

| Funder | Grant reference number | Author |
|---|---|---|
| NIDCD | R01DC018313-01A1 | Cory M Root |
| NIMH | R01MH115920-03 | Kay M Tye |
| NIMH | R37MH102441 | Kay M Tye |
| NIDDK | DP2DK102256-01S1 | Kay M Tye |
| NCCIH | DP1AT009925-04 | Kay M Tye |
| National Institutes of Health | RF1AG061831-01S1 | Paula A Desplats |
| National Institute of Mental Health | K99MH121563-02 | Fergil Mills |
| NIDCD | R00DC014516-05 | Cory M Root |
| National Defense Science and Engineering Fellowship | | James R Howe |
| CIHR Postdoctoral Fellowship | | Fergil Mills |
| Dolby Family Fund | | Kay M Tye |
| Hellman Fellowship | | Cory M Root |

The funders had no role in study design, data collection and interpretation, or the decision to submit the work for publication.

### Author contributions

James R Howe, Conceptualization, Data curation, Software, Formal analysis, Funding acquisition, Investigation, Methodology, Writing – original draft, Writing – review and editing; Chung Lung Chan, Conceptualization, Data curation, Formal analysis, Investigation, Writing – original draft; Donghyung Lee, Conceptualization, Data curation, Formal analysis, Investigation, Methodology, Writing – original draft; Marlon Blanquart, Data curation, Formal analysis, Investigation; James H Lee, Haylie K Romero, Abigail N Zadina, Investigation; Laurine Decoster, Data curation, Investigation; Mackenzie E Lemieux, Methodology; Fergil Mills, Resources, Methodology; Paula A Desplats, Conceptualization, Resources, Supervision, Investigation; Kay M Tye, Conceptualization, Resources, Supervision, Methodology, Writing – original draft; Cory M Root, Conceptualization, Resources, Data curation, Software, Formal analysis, Supervision, Funding acquisition, Validation, Investigation, Visualization, Methodology, Writing – original draft, Project administration, Writing – review and editing

### Author ORCIDs

James R Howe ⬥ https://orcid.org/0000-0002-7584-3577
Laurine Decoster ⬥ https://orcid.org/0000-0001-9889-6471
Kay M Tye ⬥ https://orcid.org/0000-0002-2435-0182
Cory M Root ⬥ https://orcid.org/0000-0003-0193-8183

## Ethics

This study was performed in strict accordance with the recommendations in the Guide for the Care and Use of Laboratory Animals of the National Institutes of Health. All of the animals were handled according to approved institutional animal care and use committee (IACUC) protocols (#S16054) of the University of California, San Diego.

Reviewer #1 (Public review): https://doi.org/10.7554/eLife.104677.3.sa1
Reviewer #2 (Public review): https://doi.org/10.7554/eLife.104677.3.sa2
Reviewer #3 (Public review): https://doi.org/10.7554/eLife.104677.3.sa3
Author response https://doi.org/10.7554/eLife.104677.3.sa4

# Additional files

## Supplementary files
MDAR checklist

Supplementary file 1. Statistical analysis details for all figures.

## Data availability

The RNA sequencing data is deposited in GEO (accession number GSE270798). The code used to analyze all data and generate all graphs can be found online at Zenodo.

The following datasets were generated:

| Author(s) | Year | Dataset title | Dataset URL | Database and Identifier |
|---|---|---|---|---|
| Howe J | 2024 | Control of innate olfactory valence by segregated cortical amygdala circuits | https://www.ncbi.nlm.nih.gov/geo/query/acc.cgi?acc=GSE270798 | NCBI Gene Expression Omnibus, GSE270798 |
| Howe J | 2024 | Control of innate olfactory valence by segregated cortical amygdala circuits | https://doi.org/10.5281/zenodo.12601490 | Zenodo, 10.5281/zenodo.12601490 |

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
