## [Editor Report · eLife Assessment]

This **fundamental** manuscript describes how the posterolateral cortical amygdala (plCoA) generates appetitive or aversive behaviors in response to odors. By combining optogenetic stimulation, single-cell RNA sequencing, and spatial analysis, the authors identify a topographically organized circuit within plCoA that governs these behaviors. The manuscript shows **convincingly** that multiple features (spatial, genetic, and projection) contribute to overall population encoding of valence. Overall, the authors conduct many challenging experiments, each of which contains the relevant controls, and the results are interpreted within the framework of their experiments.

---

## [Referee Report · Reviewer #1 (Public review)]

Summary:

This study by Howe and colleagues investigates the role of the posterolateral cortical amygdala (plCoA) in mediating innate responses to odors, specifically attraction and aversion. By combining optogenetic stimulation, single-cell RNA sequencing, and spatial analysis, the authors identify a topographically organized circuit within plCoA that governs these behaviors. They show that specific glutamatergic neurons in the anterior and posterior regions of plCoA are responsible for driving attraction and avoidance, respectively, and that these neurons project to distinct downstream regions, including the medial amygdala and nucleus accumbens, to control these responses.

Strengths:

The major strength of the study is the thoroughness of the experimental approach, which combines advanced techniques in neural manipulation and mapping with high-resolution molecular profiling. The identification of a topographically organized circuit in plCoA and the connection between molecularly defined populations and distinct behaviors is a notable contribution to understanding the neural basis of innate motivational responses. Additionally, the use of fucntional manipulations adds depth to the findings, offering valuable insights into the functionality of specific neuronal populations.

Weaknesses:

Previously described weaknesses in the study's methods and interpretation were fully addressed during revision. Locomotor behavior of the mice during head-fixed imaging experiments was added and analysis of the correlation of locomotion with neural activity was also added.

This work provides significant insights into the neural circuits underlying innate behaviors and opens new avenues for further research. The findings are particularly relevant for understanding the neural basis of motivational behaviors in response to sensory stimuli, and the methods used could be valuable for researchers studying similar circuits in other brain regions. If the authors address the methodological issues raised, this work could have a substantial impact on the field, contributing to both basic neuroscience and translational research on the neural control of behavior.

---

## [Referee Report · Reviewer #2 (Public review)]

Summary:

The manuscript by the Root laboratory and colleagues describes how the posterolateral cortical amygdala (plCoA) generates valenced behaviors. Using a suite of methods, the authors demonstrate that valence encoding is mediated by several factors, including spatial localization of neurons within the plCoA, glutamatergic markers, and projection. The manuscript shows convincingly that multiple features (spatial, genetic, and projection) contribute to overall population encoding of valence. Overall, the authors conduct many challenging experiments, each of which contains the relevant controls, and the results are interpreted within the framework of their experiments.

Strengths:

- The manuscript is well constructed, containing lots of data sets and clearly presented, in spite of the abundance of experimental results.

- The authors should be commended for their rigorous anatomical characterizations and post-hoc analysis. In the field of circuit neuroscience, this is rarely done so carefully, and when it is, often new insights are gleaned as is the case in the current manuscript.

- The combination of molecular markers, behavioral readouts and projection mapping together substantially strengthens the results.

- The focus on this relatively understudied brain region in the context is valence is well appreciated, exciting and novel.

Weaknesses:

The weaknesses noted in the primary review have all been addressed adequately.

---

## [Referee Report · Reviewer #3 (Public review)]

Summary:

Combining electrophysiological recording, circuit tracing, single cell RNAseq, and optogenetic and chemogenetic manipulation, Howe and colleagues have identified a graded division between anterior and posterior plCoA and determined the molecular characteristics that distinguish the neurons in this part of the amygdala. They demonstrate that the expression of slc17a6 is mostly restricted to the anterior plCoA whereas slc17a7 is more broadly expressed. Through both anterograde and retrograde tracing experiments, they demonstrate that the anterior plCoA neurons preferentially projected to the MEA whereas those in the posterior plCoA preferentially innervated the nucleus accumbens. Interestingly, optogenetic activation of the aplCoA drives avoidance in a spatial preference assay whereas activating the pplCoA leads to preference. The data support a model that spatially segregated and molecularly defined populations of neurons and their projection targets carry valence specific information for the odors. Moreover, the intermingling of neurons in the plCoA is consistent with prior observations. The presence of a gradient rather than a distinct separation of the cells fits the model being proposed. The discoveries represent a conceptual advance in understanding plCoA function and innate valence coding in the olfactory system.

Strengths:

The strongest evidence supporting the model comes from single-cell RNASeq, genetically facilitated anterograde and retrograde circuit tracing, and optogenetic stimulation. The evidence clear demonstrates two molecularly defined cell populations with differential projection targets. Stimulating the two populations produced opposite behavioral responses.

Weaknesses:

The weaknesses noted in primary review have all been addressed adequately.

---

## [Author Response]

The following is the authors’ response to the original reviews.

**Public Reviews:**

**Reviewer #1 (Public review):**
Summary:This study by Howe and colleagues investigates the role of the posterolateral cortical amygdala (plCoA) in mediating innate responses to odors, specifically attraction and aversion. By combining optogenetic stimulation, single-cell RNA sequencing, and spatial analysis, the authors identify a topographically organized circuit within plCoA that governs these behaviors. They show that specific glutamatergic neurons in the anterior and posterior regions of plCoA are responsible for driving attraction and avoidance, respectively, and that these neurons project to distinct downstream regions, including the medial amygdala and nucleus accumbens, to control these responses.Strengths:The major strength of the study is the thoroughness of the experimental approach, which combines advanced techniques in neural manipulation and mapping with high-resolution molecular profiling. The identification of a topographically organized circuit in plCoA and the connection between molecularly defined populations and distinct behaviors is a notable contribution to understanding the neural basis of innate motivational responses. Additionally, the use of functional manipulations adds depth to the findings, offering valuable insights into the functionality of specific neuronal populations.Weaknesses:There are some weaknesses in the study's methods and interpretation. The lack of clarity regarding the behavior of the mice during head-fixed imaging experiments raises the possibility that restricted behavior could explain the absence of valence encoding at the population level.

We agree with idea that head-fixation may alter the state of the animal and the neural encoding of odor. To address this, we have provided further analysis of walking behavior during the imaging sessions, which is provided in Figure S2. Overall, we could not identify any clear patterns in locomotor behavior that are odor-specific. Moreover, when neural activity was sorted depending on the behavioral state (walking, pausing or fleeing) we didn’t observe any apparent patterns in odor-evoked neural activity. This is now discussed in the Results and Limitations sections of the manuscript.

Furthermore, while the authors employ chemogenetic inhibition of specific pathways, the rationale for this choice over optogenetic inhibition is not fully addressed, and this could potentially affect the interpretation of the results.

The rationale was logistical. First, inhibition of over a timescale of minutes is problematic with heat generation during prolonged optical stimulation. Second, our behavioral apparatus has a narrow height between the ceiling and floor, making tethering difficult. This is now explained the results section. The trade-off of using chemogenetics is that we are silencing neurons and not specific projections. However, because we find that NAc- and MeA- projecting neurons have little shared collateralization, we believe the conclusion of divergent pathways still stands. This is now discussed in the Limitations section.

Additionally, the choice of the mplCoA for manipulation, rather than the more directly implicated anterior and posterior subregions, is not well-explained, which could undermine the conclusions drawn about the topographic organization of plCoA.

We targeted the middle region of plCoA because it contains a mixture of cell types found in both the anterior and posterior plCoA, allowing us to test the hypothesis that cell types, not intra plCoA location, elicit different responses. Had we targeted the anterior or posterior regions, we would expect to simply recapitulate the result from activation of random cells in each region. As a result, we think stimulation in the middle plCoA is a better test for the contribution of cell types. We have now clarified this in the text.

Despite these concerns, the work provides significant insights into the neural circuits underlying innate behaviors and opens new avenues for further research. The findings are particularly relevant for understanding the neural basis of motivational behaviors in response to sensory stimuli, and the methods used could be valuable for researchers studying similar circuits in other brain regions. If the authors address the methodological issues raised, this work could have a substantial impact on the field, contributing to both basic neuroscience and translational research on the neural control of behavior.
**Reviewer #2 (Public review):**
Summary:The manuscript by the Root laboratory and colleagues describes how the posterolateral cortical amygdala (plCoA) generates valenced behaviors. Using a suite of methods, the authors demonstrate that valence encoding is mediated by several factors, including spatial localization of neurons within the plCoA, glutamatergic markers, and projection. The manuscript shows convincingly that multiple features (spatial, genetic, and projection) contribute to overall population encoding of valence. Overall, the authors conduct many challenging experiments, each of which contains the relevant controls, and the results are interpreted within the framework of their experiments.Strengths:- For a first submission the manuscript is well constructed, containing lots of data sets and clearly presented, in spite of the abundance of experimental results.- The authors should be commended for their rigorous anatomical characterizations and posthoc analysis. In the field of circuit neuroscience, this is rarely done so carefully, and when it is, often new insights are gleaned as is the case in the current manuscript.- The combination of molecular markers, behavioral readouts and projection mapping together substantially strengthen the results.- The focus on this relatively understudied brain region in the context is valence is well appreciated, exciting and novel.Weaknesses:- Interpretation of calcium imaging data is very limited and requires additional analysis and behavioral responses specific to odors should be considered. If there are neural responses behavioral epochs and responses to those neuronal responses should be displayed and analyzed.

We have now considered this, see response above.

- The effect of odor habituation is not considered.

We considered this, but we did not find any apparent differences in valence encoding as measured by the proportion of neurons with significant valence scores across trials (see Figure 1J).

- Optogenetic data in the two subregions relies on very careful viral spread and fiber placement. The current anatomy results provided should be clear about the spread of virus in A-P, and D-V axis, providing coordinates for this, to ensure readers the specificity of each sub-zone is real.

We were careful to exclude animals for improper targeting. The spread of virus is detailed in Figures S3, S8 & S9.

- The choice of behavioral assays across the two regions doesn't seem balanced and would benefit from more congruency.

The choice of the 4-quadrant assay was used because this study builds off of our prior experiments that demonstrate a role for the plCoA in innate behavior. It is noteworthy that the responses to odor seen in this assay are generally in agreement with other olfactory behavioral assays, so one wouldn’t predict a different result. Moreover, the approach and avoidance responses measured in this assay are precisely the behaviors we wish to understand. We did examine other non-olfactory behavioral readouts (Figures S3, S8), and didn’t observe any effect of manipulation of these pathways.

- Rationale for some of the choices of photo-stimulation experiment parameters isn't well defined.

The parameters for photo-stimulation were based on those used in our past work (Root et al., 2014). We used a gradient of frequency from 1-10 Hz based on the idea that odor likely exists in a gradient and this was meant to mimic a potential gradient, though we don’t know if it exists. The range in stimulation frequencies appears to align with the actual rate of firing of plCoA neurons (Iurilli et al., 2017).

**Reviewer #3 (Public review):**
Summary:Combining electrophysiological recording, circuit tracing, single cell RNAseq, and optogenetic and chemogenetic manipulation, Howe and colleagues have identified a graded division between anterior and posterior plCoA and determined the molecular characteristics that distinguish the neurons in this part of the amygdala. They demonstrate that the expression of slc17a6 is mostly restricted to the anterior plCoA whereas slc17a7 is more broadly expressed. Through both anterograde and retrograde tracing experiments, they demonstrate that the anterior plCoA neurons preferentially projected to the MEA whereas those in the posterior plCoA preferentially innervated the nucleus accumbens. Interestingly, optogenetic activation of the aplCoA drives avoidance in a spatial preference assay whereas activating the pplCoA leads to preference. The data support a model that spatially segregated and molecularly defined populations of neurons and their projection targets carry valence specific information for the odors. The discoveries represent a conceptual advance in understanding plCoA function and innate valence coding in the olfactory system.Strengths:The strongest evidence supporting the model comes from single cell RNASeq, genetically facilitated anterograde and retrograde circuit tracing, and optogenetic stimulation. The evidence clear demonstrates two molecularly defined cell populations with differential projection targets. Stimulating the two populations produced opposite behavioral responses.Weaknesses:There are a couple of inconsistencies that may be addressed by additional experiments and careful interpretation of the data.Stimulating aplCoA or slc17a6 neurons results in spatial avoidance, and stimulating pplCoA or slc17a7 neurons drives approach behaviors. On the other hand, the authors and others in the field also show that there is no apparent spatial bias in odor-driven responses associated with odor valence. This discrepancy may be addressed better. A possibility is that odor-evoked responses are recorded from populations outside of those defined by slc17a6/a7. This may be addressed by marking activated cells and identifying their molecular markers. A second possibility is that optogenetic stimulation activates a broad set of neurons that and does not recapitulate the sparseness of odor responses. It is not known whether sparsely activation by optogenetic stimulation can still drive approach of avoidance behaviors.

We agree that marking specific genetic or projection defined neurons could help to clarify if there are some neurons have more selective valence responses. However, we are not able to perform these experiments at the moment. We have included new data demonstrating that sparser optogenetic activation evokes behaviors similar in magnitude as the broader activation (see Figure S4).

The authors show that inhibiting slc17a7 neurons blocks approaching behaviors toward 2-PE. Consistent with this result, inhibiting NAc projection neurons also inhibits approach responses. However, inhibiting aplCOA or slc17a6 neurons does not reduce aversive response to TMT, but blocking MEA projection neurons does. The latter two pieces of evidence are not consistent with each other. One possibility is that the MEA projecting neurons may not be expressing slc17a6. It is not clear that the retrogradely labeling experiments what percentage of MEA- and NACprojecting neurons express slc17a6 and slc17a7. It is possible that neurons expressing neither VGluT1 nor VGluT2 could drive aversive or appetitive responses. This possibility may also explain that silencing slc17a6 neurons does not block avoidance.

We have now performed RNAscope staining on retrograde tracing to better define this relationship. Although the VGluT1 and VGluT2 neurons have biased projections to the MeA and NAc, respectively, there is some nuance detailed in Figure S10. Generally, MeA projecting neurons are predominately VGluT2+, whereas NAc projecting have about 20% that express both. Some (less than 35%) retrogradely labeled neurons were not detected as VGluT1 or VGluT2 positive, suggesting that other populations could also contribute. We agree that the discrepancy between MeA-projection and VGluT2 silencing is likely due to incomplete targeting of the MeA-projecting population with the VGluT2-cre line. This is included in the Discussion section.

**Recommendations for the authors:**

**Reviewer #1 (Recommendations for the authors):**
Main:(1) For the head-fixed imaging experiments, what is the behavior of the mice during odor exposure? Could the weak reliability of individual neurons be due to a lack of approach or avoidance behavior? Could restricted behavior also explain the lack of valence encoding at the population level?

We agree that this is a limitation of head-fixed recordings. In the revised manuscript we did attempt to characterize their behavioral response, and look for correlations in odor representation. Although we did find different patterns of odor-evoked walking behavior, these patterns were not reliable or specific to particular odors (Figure S2). For example, one might expect aversive odors to pause walking or elicit a fast fleeing-like response, but we did not observe any apparent differences for locomotion between odors as all odors evoked a mixture of responses (Figure S2A-D, text lines 208-232). We then examined responses to odor depending on the behavioral state (walking, pausing or fleeing) and didn’t observe any apparent patterns in odor responses (Figure S2E,F). Lastly, we acknowledge in the text that the lack of valence encoding may be an artifact of head-fixation (see lines 849-857).

(2) For the optogenetic manipulations of Vglut1 and Vglut2 neurons, why was the injection and fiber targeted to the medial portion of the plCoA, if the hypothesis was that these glutamatergic neuron populations in different regions (anterior or posterior) are responsible for approach and avoidance?

We targeted the middle region of plCoA because it contains a mixture of cell types found in both the anterior and posterior plCoA, allowing us to test the hypothesis that cell types, not intraplCoA location, elicit different responses. Had we targeted the anterior or posterior regions, we would expect to simply recapitulate the result from activation of random cells in each region. As a result, we think stimulation in the middle plCoA is a better test for the contribution of cell types. We have clarified this in the text (Lines 417-419).

Could this explain the lack of necessity with the DREADD experiments?

For the loss of function experiments, a larger volume of virus was injected to cover a larger area and we did confirm targeting of the appropriate areas. Though, it is always possible that the lack of necessity is due to incomplete silencing.

Further, why was an optogenetic inhibition approach not utilized?

Although optogenetic inhibition could have plausibly been used instead, we chose chemogenetic inhibition for two reasons: First, for minutes-long periods of inhibition, optical illumination poses the risk of introducing heat related effects (Owen et al., 2019). In fact, we first tried optical inhibition but controls were exhibited unusually large variance. Second, it is more feasible in our assay as it has a narrow height between the floor and lid that complicates tethering to an optic fiber. Past experiments overcame this with a motorized fiber retraction system (Root et al., 2014), but this is highly variable with user-dependent effects, so we found chemogenetics to be a more practical strategy. We have added a sentence to explain the rationale (see lines 561-563).

(3) The specific subregion of the nucleus accumbens that was targeted should be named, as distinct parts of the nucleus accumbens can have very different functions.

We attempted to define specific subregions of the nucleus accumbens and found that plCoA projection is not specific to the shell or core, anterior or posterior, rather it broadly innervates the entire structure. We have added a note about this in manuscript (see lines 470-471). Given that we did not find notable subregion-specific outputs within the NAc, targeting was directed to the middle region of NAc, with coordinates stated in the methods.

(4) Why was an intersectional DREADD approach used to inhibit the projection pathways, as opposed to optogenetic inhibition? The DREADD approach could potentially affect all projection targets, and the authors might want to address how this could influence the interpretation of the results.

This is partly addressed above in point 2. As for interpretation, we acknowledge that the intersectional approach silences the neurons projecting to a given target and not the specific projection and we have been careful with the wording. Although this may complicate the conclusion, we did map the collaterals for NAc and MeA projecting neurons and find that neurons do not appreciably project to both targets and have minimal projections to other targets. We have now taken care to state that we silence the neurons projecting to a structure, not silencing the projection, and we acknowledge this caveat. However, since the MeA- and NAcprojecting neurons appear to be distinct from each other (largely not collateralizing to each other), the conclusion that these divergent pathways are required still stands. We have added discussion of this in the Limitations section (see lines 859-863).

Minor:(1) Line 402 needs a reference.

We have added the missing reference (now line 441).

(2) The Supplemental Figure labeling in the main text should be checked carefully.

Thank you for pointing this out. We have fixed the prior errors.

(3) Panel letter D is missing from Figure 2.

This has been fixed.

**Reviewer #2 (Recommendations for the authors):**
Major Concerns, additional experiments:- In the calcium imaging experiments mice were presented with the same odor many times. Overall responses to odor presentations were quite variable and appear to habituate dramatically (Figure S1F). The general conclusion from these experiments are a lack of consistent valence-specific responses of individual neurons, but I wonder if this conclusion is slightly premature. A few potential explanatory factors that may need additional attention are: -First, despite recording video of the mouse's face during experiments, no behavioral response to any odor is described. Is it possible these odors when presented in head-fixed conditions do not have the same valence?

Yes, we agree that this is a possibility. We have added a discussion in the Limitations section (see lines 849-857). We have also added additional behavioral analysis discussed below.

On trials with neural responses are there behavioral responses that could be quantified?

We have now added data in which we attempt to characterize their behavioral response, to look for correlations in odor representation (see lines 208-228). Although we did observe different patterns of odor-evoked walking behavior, these patterns were not reliable or specific to particular odors (Figure S2). One might expect aversive odors to pause walking or elicit a fast fleeing-like response, but we did not observe any apparent differences for locomotion between odors (Figure S2A-D). Next, we examined responses to odor depending on the behavioral state (walking, pausing or fleeing) and didn’t observe any meaningful differences in odor responses (Figure S2E,F). Lastly, we acknowledge that the odor representation may be different in freely moving animals that exhibit dynamic responses to odor (see lines 859-857).

- Habituation seems to play a prominent role in the neural signals, is there a larger contribution of valence if you look only at the first delivery (or some subset of the 20 presentations) of an odor type for a given trial?

Indeed, we considered this, but we did not find any apparent differences in valence encoding as measured by the proportion of neurons with significant valence scores across trials (see Figure 1J).

- Is it reasonable to exclude valence encoding as a possibility when largely neurons were unresponsive to the positive valence odors (2PE and peanut) chosen when looking at the average cluster response (Figure 1F)?

It is true that we see fewer neurons responding to the appetitive odors (Figure 1H) and smaller average responses within the cluster, but some neurons do respond robustly. If these were valence responses, we would predict that neural responses should be similarly selective, but we do not observe any such selectivity. The sparseness of responses to appetitive odors does cause the average cluster analysis (Figure 1F) to show muted responses to these odors, consistent with the decreased responsivity to appetitive odors. Moreover, single neuron response analysis reveals that a given neuron is not more likely to respond to appetitive or aversive odors with any selectivity greater than chance. For these reasons, we think it is reasonable to conclude an absence of valence responses, which is consistent with the conclusion from another report (Iurilli et al., 2017).

- While the preference and aversion assay with 4 corners is an interesting set-up and provides a lot of data for this particular manuscript. It would be helpful to test additional behaviors to determine whether these circuits are more conserved. As it stands the current manuscript relies on very broad claims using a single behavioral readout. Some attempts to use head-fixed approaches with more defined odor delivery timelines and/or additional valenced behavioral readouts is warranted.

We appreciate the suggestion, but are not able to perform these experiments at the moment. The choice of the 4-quadrant assay was used because it built off of our prior experiments that demonstrate a role for the plCoA in innate behavior. It is noteworthy that the responses to odor seen in this assay are generally in agreement with other olfactory behavioral assays, so one wouldn’t predict a different result. The approach and avoidance responses measured in this assay are precisely the behaviors we wish to understand. Moreover, we did examine other nonolfactory behavioral readouts (Figures S3, S8), and didn’t observe any effect of manipulation of these pathways. Lastly, we have tried to define parameters for head-fixed behavior that would permit correlation of neural responses with behavior, including longer stimulations and closed loop locomotion control of odor concentration, but were unsuccessful at establishing parameters that generated reliable behavioral responses. We acknowledge that one limitation of the study is the limited behavioral tests with two odors and whether the circuits are more broadly necessary for other odors.

Minor comments:• Please define PID in the Results when it is first introduced.

Done (see line 154)

• Line 412 Figure S5C-N should be Figure S6C-N.

Fixed. Now Figure S8C-N due to additional figures (see line 451).

• Throughout the Discussion it would be helpful if the authors referred to specific Figure panels that support their statements e.g. lines 654-656 "[...] which is supported by other findings presented here showing that both VGluT2+ and VGluT1+ neurons project to MeA, while the projection to NAc is almost entirely composed of VGluT1+ neurons".

Thank you for the suggestion. We have added figure references in the discussion.

• Line 778 "producing" should be "produce".

Corrected (see line 840)

• The figures are very busy, especially all the manipulations. The authors are commended for including each data point, but they might consider a more subtle design (translucent lines only for each animal, and one mean dot for the SEM), just to reduce the overall clutter of an already overwhelming figure set. But this is ultimately left to the authors to resolve and style to their liking.

Thank you for the suggestion. We have tried some different styles but like the original best.

**Reviewer #3 (Recommendations for the authors):**
If within reach, I suggest that the author determine the percentage of retrogradely labeled neurons to NAc or MEA that expresses GluT1 and GluT2.

We have done this for the middle region plCoA that has the greatest mixture of cell types (See Figure S10, lines 504-517). We find that the MeA projecting neurons are mostly *VGluT2*+ with a minority that express both *VGluT1* and *VGlut2*. NAc-projecting neurons are primarily *VGluT1*+ with about 20% expressing *VGlut2* as well.

It would also be nice to sparse label of aplCoA and pplCoA using ChR2 to see if sparse activation drives approach or avoidance.

We agree that it would be useful to vary the sparseness of the ChR2 expression, to see if produces similar results. We examined this using sparsely labeled odor ensembles, as previously done (Root et al., 2014). Briefly, we used the Arc-CreER mouse to label TMT responsive neurons with a cre-dependent ChR2 AAV vector targeted to the anterior or posterior regions, while previously we had broadly targeted the entirety of plCoA. We had established that this labeling method captures about half of the active cells detected by *Arc* expression, which is on the order of hundreds of neurons rather than thousands by broad cre-independent expression. Remarkably, we get effects similar in magnitude that are not significantly different from that with broader activation of the anterior or posterior domains (see new Figure S4, lines 267-288). It still remains possible that there is a threshold number of neurons that are necessary to elicit behavior, but that is beyond the scope of the current study. However, these data indicate that the effect of activating anterior and posterior domains is not an artifact of broad stimulation.